# The Prognostic Significance of *FKBP1A* and Its Related Immune Infiltration in Liver Hepatocellular Carcinoma

**DOI:** 10.3390/ijms232112797

**Published:** 2022-10-24

**Authors:** Zhongguang Li, Ying Cui, Qinchun Duan, Jianfei Zhang, Danyang Shao, Xixi Cao, Yuru Gao, Shulin Wang, Jiali Li, Odell D. Jones, Xinjuan Lei, Liyang Wang, Xin Zhou, Mengmeng Xu, Jianjie Ma, Yingli Liu, Xuehong Xu

**Affiliations:** 1Laboratory of Cell Biology, Genetics and Developmental Biology, Shaanxi Normal University College of Life Sciences, Xi’an 710062, China; 2Department of Internal Medicine, University Hospital Shaanxi Normal University, Xi’an 710062, China; 3University Laboratory Animal Resources (ULAR), University of Pennsylvania School of Medicine, Philadelphia, PA 19144, USA; 4Department of Pediatrics, Columbia University, New York, NY 10032, USA; 5Department of Surgery, Davis Heart and Lung Research Institute, Ohio State University School of Medicine, Columbus, OH 43210, USA

**Keywords:** *FKBP1A*, hepatocellular carcinoma, bioinformatics analysis, immune infiltration, prognostic biomarker

## Abstract

Liver hepatocellular carcinoma (LIHC) remains a global health challenge with poor prognosis and high mortality. *FKBP1A* was first discovered as a receptor for the immunosuppressant drug FK506 in immune cells and is critical for various tumors and cancers. However, the relationships between *FKBP1A* expression, cellular distribution, tumor immunity, and prognosis in LIHC remain unclear. Here, we investigated the expression level of *FKBP1A* and its prognostic value in LIHC via multiple datasets including ONCOMINE, TIMER, GEPIA, UALCAN, HCCDB, Kaplan–Meier plotter, LinkedOmics, and STRING. Human liver tissue microarray was employed to analyze the characteristics of FKBP1A protein including the expression level and pathological alteration in cellular distribution. *FKBP1A* expression was significantly higher in LIHC and correlated with tumor stage, grade and metastasis. The expression level of the FKBP1A protein was also increased in LIHC patients along with its accumulation in endoplasmic reticulum (ER). High *FKBP1A* expression was correlated with a poor survival rate in LIHC patients. The analysis of gene co-expression and the regulatory pathway network suggested that *FKBP1A* is mainly involved in protein synthesis, metabolism and the immune-related pathway. *FKBP1A* expression had a significantly positive association with the infiltration of hematopoietic immune cells including B cells, CD8^+^ T cells, CD4^+^ T cells, macrophages, neutrophils, and dendritic cells. Moreover, M2 macrophage infiltration was especially associated with a poor survival prognosis in LIHC. Furthermore, *FKBP1A* expression was significantly positively correlated with the expression of markers of M2 macrophages and immune checkpoint proteins such as *PD-L1*, *CTLA-4*, *LAG3* and *HAVCR2*. Our study demonstrated that *FKBP1A* could be a potential prognostic target involved in tumor immune cell infiltration in LIHC.

## 1. Introduction

Primary liver cancer (PLC) is ranked as the sixth most common cancer in the world and the third leading cause of cancer death in 2020, with approximately 906,000 new cases and 830,000 deaths [1]. Liver hepatocellular carcinoma (LIHC) is the commonly diagnosed PLC type, which accounts for 75–85% of all PLC cases [1,2]. Currently, the critical issue for LIHC is the poor five-year overall survival rate of only 30–40% including that diagnosed at an early stage [3]. The clinical treatment for LIHC has been greatly developed, including chemotherapy, radiotherapy, transarterial chemoembolization, surgical resection, and liver transplantation surgery. However, these treatments show less satisfactory efficacy [4]. In recent years, as the immune system plays an important part in controlling LIHC progression, the immunotherapy for LIHC has grown dramatically and changed the treatment paradigm [5]. Thus, identifying novel therapeutic biomarkers associated with the immune microenvironment is critical for new immunotherapeutic treatment for LIHC, which could improve the prognosis of patients.

As a highly conserved immunophilin family, FK506 binding proteins (FKBPs) play an important role in the regulation of adaptive immune response, inflammation, cancer, heart disease and neurodegeneration [6,7,8]. Immunosuppressant drugs FK506 and rapamycin bind to FKBP12 leading to inhibition of the PPIase activity [8]. The rapamycin-FKBP12 complex specifically binds to mTOR (mammalian target of rapamycin) and blocks its serine/threonine kinase function, which subsequently inhibits the cytokine-stimulated protein synthesis, cell growth and proliferation [9]. FKBP12 directly interacts with Ca^2+^ release channel ryanodine receptor 1 (RyR1) or the inositol 1,4,5-trisphosphate receptor (IP3R) to regulate intracellular Ca^2+^ homeostasis [10,11]. In addition, FKBP12 also interacts with type I receptor (TGF-βR1) and plays a role in TGF-beta mediated signaling in both normal cells and tumors [7]. Although there are reports showing that FKBP12 can stimulate oncoprotein mouse double minute 2 (MDM2) self-ubiquitination and degradation to enhance the sensitivity in breast cancer cells and chemotherapy [12,13], its prognostic value and association with tumor immunity in LIHC have not been clearly defined.

As a member of the immunophilin superfamily, FKBP1A can bind to immunosuppressive drugs including FK506, rapamycin and cyclosporin A (CsA) and interacts in the mTOR pathway. FKBP1A was first identified in dendrite cells of peripheral blood cells and its related molecular mechanism has been under investigation for decades. One FKBP1A molecule binds to an RyR unit of RyR tetramer, and the RyR–FKBP1A complex is in charge of excitation–contraction coupling in muscle, and excitation–conduction contractile coupling in neuronal cells via the calcium-induced calcium release (CICR) signaling system [14,15,16]. The FKBP1A-RyR complex medicated CICR plays an important role in multiple cell biological processes in normal physiological and pathological cells.

In recent years, the emergence of immunotherapy has brought strong hope for the treatment of liver cancer. The application of immune checkpoint inhibitors (ICIs), especially programmed cell death protein 1 (PD-1), programmed cell death ligand 1 (PD-L1), and cytotoxic T lymphocyte antigen 4 (CTLA-4) has shown great promise and critical progress for LIHC treatment [17]. However, many cancer patients fail to respond to immune checkpoint blockade [18]. Basic research and clinical trials exploring the biomarkers of immunotherapy to predict efficacy are still limited, and it is not certain which biomarkers can effectively examine the efficacy of immunotherapy [18]. Our study on FKBP1A and LIHC may provide competent possibilities for pathological events. 

Multiple studies reported that *FKBP1A* was involved in tumorigenesis including head and neck squamous cell carcinoma [19], prostatic cancer cell line [20], lung adenocarcinoma [21] and breast cancer [12]. In the immune system, FKBP1A directly led to mTOR activity via rapamycin-recruited SIRT2 for its deacetylation [22]. Recently, miR-195-5p/FKBP1A was identified as a key factor for paclitaxel-resistant prostate cancer cells to paclitaxel via LncRNA AFAP1-AS binding to the 3ʹUTR of *FKBP1A* [20].

In the present work, we analyzed the *FKBP1A* expression and its relationship with prognosis in the tumorigenesis and progression in patient chips compared to bioinformatics analysis. Our data revealed that FKBP1A expression was significantly upregulated in LIHC compared with normal liver tissues by bioinformatics and immunohistochemical analysis with human liver tissue microarray. It is certain that the patients with a high expression of *FKBP1A* showed poor prognosis. The *FKBP1A* upregulation is significantly associated with more immune cell infiltration in LIHC, which indicates that *FKBP1A* could be a potential biomarker for the diagnosis and prognosis of LIHC.

## 2. Results

While tumor immune checkpoint inhibitors have potential in the treatment of LIHC, only a small percentage of LIHC patients with advanced hepatocellular carcinoma at a high risk of death achieved the desired results from current conventional immunotherapy. We discovered in this study that the immunophilin family member *FKBP1A* is widely expressed in the LIHC tissues and increased along with the grading development. A comprehensive understanding of the role of *FKBP1A* in human LIHC could provide critical information for a breakthrough in the development of the current conventional immunotherapy.

### 2.1. Differential Expression of FKBP1A in Different Cancers

We first analyzed the mRNA expression level of *FKBP1A* in multiple tumors and adjacent normal tissue types based on the data from the Oncomine database. The results revealed that the expression of *FKBP1A* mRNA from liver cancer increased in three data sets compared to the normal tissues (Figure 1A). Meanwhile, the expression levels of *FKBP1A* mRNA were higher in bladder cancer, gastric cancer, head and neck cancer, kidney cancer, lymphoma, myeloma, ovarian cancer, prostate cancer and sarcoma compared to the normal tissues. However, lower expression was detected in colorectal cancer and esophageal cancer (Figure 1A). These results suggested that the expression pattern of *FKBP1A* is different in most human cancers.

To confirm this differential expression pattern of *FKBP1A* mRNA in human cancers, we further examined the *FKBP1A* expression from TIMER2.0. It showed that *FKBP1A* mRNA expression was significantly higher in most tumor tissues, including LIHC, BLCA (bladder urothelial carcinoma), CHOL (cholangiocarcinoma), ESCA (esophageal carcinoma), HNSC (head and neck squamous cell carcinoma), KIRC (kidney renal clear cell carcinoma), KIRP (kidney renal papillary cell carcinoma) and STAD (stomach adenocarcinoma) (*p* < 0.05, Figure 1B). In addition, *FKBP1A* expression was significantly lower in COAD (colon adenocarcinoma), KICH (kidney chromophobe), LUAD (lung adenocarcinoma), THCA (thyroid carcinoma) (*p* < 0.05, Figure 1B). 

Based on the Oncomine and TIMER2.0 databases, both analyses present that the significantly higher expression may tightly associated LIHC tumorigenesis while there is a differential expression pattern of *FKBP1A* in the different cancers listed above.

### 2.2. Expression of FKBP1A Is Upregulated in Liver Hepatocellular Carcinoma

In order to confirm incidence of the high expression level of *FKBP1A* in LIHC, we therefore analyzed the data from GEPIA and UALCAN databases as well in multiple datasets with the HCCDB database. Using the GEPIA database, we confirmed that the *FKBP1A* expression significantly increased in LIHC (*p* < 0.05, Figure 2A). The analysis showed that *FKBP1A* expression in LIHC patients was significantly increased compared with adjacent/healthy tissues in 10 of 12 datasets from HCCDB database (Appendix A). Focusing on the relationship between *FKBP1A* expression level and the clinical characteristics of patients with LIHC, we examined possible subgroup associations with *FKBP1A* expression based on tumor stage, grade, and lymph node metastasis through the UALCAN database. The results showed that *FKBP1A* expression level was significantly higher in the advanced tumors by grade analysis (*p* < 0.05, Figure 2B). The *FKBP1A* was significantly upregulated in stages 1, 2, 3, and 4 compared with normal liver tissues (*p* < 0.0001, Figure 2C).

Using UALCAN databases, we further examined the relationship between the *FKBP1A* expression levels and the lymph node metastasis. The analysis showed that the expression of *FKBP1A* was significantly increased in N0 (no regional lymph node metastasis, *p* < 0.0001, Figure 2D) and N1 (metastases in 1 to 3 axillary lymph nodes, *p* < 0.01, Figure 2D). Taken together, these results suggest that higher *FKBP1A* expression is broadly associated with the development of LIHC along with metastasis into lymph nodes. We would like to claim that the upregulated *FKBP1A* expression in LIHC might act as a potential diagnostic indicator for detecting LIHC development.

### 2.3. Upregulated FKBP1A Protein Is Accumulated in Endoplasmic Reticulum in Liver Hepatocellular Carcinoma

In order to identify the FKBP1A protein distribution to characterize its cellular functional disorder, we further applied immunohistochemistry (IHC) assays to evaluate the protein levels and cellular localization of FKBP1A in human liver tissue microarray (Figure 3A). We compared the FKBP1A protein expression in the tissues of healthy liver, adjacent to LIHC and LIHC. In line with the transcriptomics data, FKBP1A staining was significantly increased in tumor tissues compared with normal liver tissues (grouped by combining samples of healthy individuals, and tumor-adjacent tissues) (*p* < 0.001, Figure 3A–C). It was confirmed that the protein expression in LIHC is significantly higher than that in healthy liver, and adjacent to LIHC, respectively, with no significant difference in healthy livers and LIHC (Figure 3A–C).

In addition, we also performed immunofluorescence staining for the endoplasmic reticulum (ER) marker ryanodine receptor (RyR) and co-localization analysis of FKBP1A with RyR (Figure 4A,B). Immunofluorescence staining analysis showed that the expression of FKBP1A was mainly distributed in the cytoplasm of normal and tumorigenic hepatocytes (Figure 4A). The expression of FKBP1A was homogeneously distributed in normal liver tissues. In the hepatic cells, the protein staining was distributed in the cytoplasm located in the network of the endoplasmic reticulum. The staining signal presented evenly in the entire cytosol of the cell in the normal tissues (Figure 4A). However, in LIHC tissues, although the FKBP1A protein staining still mainly distributes in the cytoplasmic network ER of LIHC hepatic cells with less staining in the cytosol (Figure 4A,C), the ER appeared significantly enlarged and the protein concentrated in the ER of hepatic cells in LIHC patients (Figure 4A,C).

The cytosol expression of FKBP1A showed no significant differences between healthy liver, adjacent to LIHC and LIHC. However, the ER expression of FKBP1A presented strong significant differences comparing LIHC ER to healthy liver ER and ER in tissue adjacent to LIHC (Figure 4C). 

### 2.4. Intracellular Abnormality of FKBP1A Expression Distribution Is Accompanied by ER-Accumulation during Developing Severity of Hepatocellular Carcinoma

Interestingly, our immune-histochemical analysis of the association of FKBP1A expression also unveiled a significant alteration in tumor cellular distribution and the accumulation of FKBP1A protein in ER. In normal hepatic cells, the FKBP1A expression is evenly distributed in the cytosolic matrix without significant cellular distribution or slightly uneven differences (Figure 3A(a,c,e),B and Figure 4A(a,b,c)). However, during the development of LIHC, the cellular distribution in hepatic cells becomes uneven within the cytosolic matrix and concentrates on peri-nuclear localization. Along with the cytosolic increase in FKBP1A expression during growing severity, more and more FKBP1A protein accumulated within the LIHC ER condensing in peri-nuclear localization from normal hepatic cells to LIHC (Figure 3A(b,d,f) and Figure 4A(d,e,f),C). As the cellular FKBP1A protein is recognized as the regulator of ER RyR and IP3R, it is certain that this intra-ER accumulation of the FKBP1A (Figure 4A–C) will extinguish its cellular function, targeting RyR- and IP3R-mediated intracellular calcium pathways.

In addition, our immune-histochemical analysis of the association of FKBP1A expression with tumor-node-metastasis (TNM) stage and tumor histologic grade in patients with LIHC unveiled a significant increase along with the developing severity of LIHC. The FKBP1A protein expression was significantly upregulated in T2N0M0 or T3N0M0 tumors (Figure 5A,C) and grade 2 or 3 tumors (Figure 5B,D) compared with normal liver tissues. There was no significant difference in FKBP1A protein expression in advanced tumors compared to early-stage tumors (*p* > 0.05, Figure 5C). However, the relative expression of FKBP1A protein was higher at tumor grade 2 and 3 than that at grade 1 (Figure 5D). Moreover, the FKBP1A protein expression in ER was also significantly upregulated in T2N0M0 or T3N0M0 tumors (Figure 5E) and grade 2 or 3 tumors (Figure 5F) compared with normal liver tissues. The distribution of FKBP1A expression in the ER also increased significantly with the increase in tumor grade (Figure 5F). These results indicate that increased FKBP1A level is associated with severity in the development of LIHC, which indicates its key role in LIHC tumorigenesis.

### 2.5. FKBP1A Expression Could Be Critical Factor for the Prognosis of LIHC Patients

To enhance the understanding of the correlation for the potential mechanism of *FKBP1A* gene expression in liver cancer, we further analyzed the correlation between *FKBP1A* expression and various clinical characteristics in LIHC patients using the Kaplan–Meier plotter. The analysis revealed that the overall survival (Figure 6A, HR = 2.11, 95% CI = 1.44 to 3.08, Logrank *p* = 8.3 × 10^−5^), relapse-free survival (Figure 6B, HR = 2.01, 95% CI = 1.39 to 2.89, Logrank *p* = 0.00014), progression-free survival (Figure 6C, HR = 1.95, 95% CI = 1.37 to 2.76, Logrank *p* =0.00014), and disease-specific survival (Figure 6D, HR = 2.36, 95% CI = 1.44 to 3.86, Logrank *p* = 0.00046) were significantly reduced when the expression level of *FKBP1A* was high in the patients.

The high expression of *FKBP1A* corresponded with worse overall survival and progression-free survival regardless of gender (female and male) and race (white and Asian) (Table 1, *p* < 0.05). The high *FKBP1A* expression was significantly associated with a poor overall survival in stage 1/3, grade 1/2/3 and AJCC-T 1/3 in LIHC patients (Table 1, *p* < 0.05) but not stage 2 and AJCC-T 2 (Table 1, *p* > 0.05). Meanwhile, the high expression was significantly associated with a poor progression-free survival in stage 1/3, grade 1/2 and AJCC-T 1/3 in LIHC patients (Table 1, *p* < 0.05) but not stage 2, grade 3 and AJCC-T 2 (Table 1, *p* > 0.05). Specifically, the high *FKBP1A* expression correlated with worse overall survival in LIHC patients without vascular invasion, and that in patients regardless of alcohol consumption or viral hepatitis (Table 1, *p* < 0.05). Interestingly, *FKBP1A* expression was only associated with progression-free survival in the absence of the hepatitis virus or alcohol consumption (Table 1, *p* < 0.05), but not in the presence of the hepatitis virus or alcohol consumption (Table 1, *p* > 0.05). These differences in clinical features suggest that the application of *FKBP1A* as an indicator should be considered in the above conditions of HILC patients. 

To further confirm the prognostic potential of *FKBP1A* in LIHC patients, we used the GEPIA database employing transcriptomic sequencing data in TCGA to evaluate the impact of *FKBP1A* expression on overall survival and disease-free survival rates. The Cox P/log-rank *p* value and hazard ratio with 95% intervals were calculated. We set Cox P/log-rank *p* = 0.05 as the threshold. The patients were divided into two groups based on the median level of *FKBP1A* expression in each queue. The results showed that a higher expression level of *FKBP1A* was correlated with a worse prognosis of overall survival (Figure 6E, *p* = 0.0023) and disease-free survival (Figure 6F, *p* = 0.0019). Based on the analysis above, we claim that LIHC patients with high *FKBP1A* expression usually have poor survival outcomes, which suggests that the high *FKBP1A* expression may be a critical risk factor for a poor prognosis in LIHC patients.

To determine the risk factors related with LIHC overall survival, we used both univariate and multivariate Cox regression to carry out the analysis. Univariate Cox analysis identified the potential OS-related variables on *FKBP1A* and others including the age, gender, pT stage, pTNM stage and grade. The univariate (hazard ratio, 1.766; 95% CI, 1.375–2.268; *p* < 0.001; Table 2) and multivariate (hazard ratio, 1.546; 95% CI, 1.185–2.017; *p* = 0.001; Table 2) Cox analyses showed that *FKBP1A* was an independent prognostic risk factor for LIHC overall survival (Table 2).

### 2.6. The Co-Expression Networks of FKBP1A Directs to Immunodeficiency Pathway in LIHC 

To discovery the biological comprehension of *FKBP1A* in the LIHC cohort, the “LinkFinder” module in LinkedOmics was employed to check the co-expression patterns of *FKBP1A* and its related pathways, which could provide critical key points to broaden the possible use for the phenomenon of *FKBP1A*-associated upregulation in LIHC tumorigenesis.

As shown in the volcano plot (Figure 7A), 7206 genes (dark red dots) showed significant positive correlations with *FKBP1A*, whereas 5234 genes (dark green dots) showed significant negative correlations (false discovery rate [FDR] < 0.01). The three most significant genes positively associated with *FKBP1A* expression were *TMSB10* (positive rank #1, r = 0.7252, *p* = 8.844 × 10^−62^), *IPTA* (r = 0.704, *p* = 8.577 × 10^−57^) and *SNRPB2* (r = 0.68, *p* = 1.15 × 10^−51^) (Appendix A). The other three most significant negatively associated genes were *MYO18A* (negative rank #1, r = −0.6654, *p* = 1.307 × 10^−48^), *NFIC* (r = −0.6425, *p* = 1.428 × 10^−44^) and *TOM1L1* (r = −0.6422, *p* = −1.569 × 10^−44^) (Appendix A). The top 50 significant genes positively and negatively correlated with *FKBP1A* are also shown in the heat map (Appendix A).

Therefore, to inspect the functional network of *FKBP1A* co-expression genes on LIHC development, we used GSEA to analyze the data from LinkedOmics, and examined the results of the *FKBP1A*-associated GSEA enrichment data with “affinity propagation”. The Gene Ontology (GO) analysis for cellular components (CC) showed that genes co-expressed with *FKBP1A* were mainly involved in subcellular fractions such as ribosomes, mitochondria, and endoplasmic reticulum (Appendix A). The biological processes (BP) and molecular function (MF) ontologies showed that *FKBP1A* co-expressed genes were mainly enriched in protein synthesis and cell metabolic processes (Appendix A). The important data present that *FKBP1A* co-expressed genes were also associated with immunology pathways including the response to interleukin 12, granulocyte activation, and NF-kappaB binding (Appendix A). Moreover, the Kyoto Encyclopedia of Genes and Genomes (KEGG) pathway analysis indicated that the function of genes co-expressed with *FKBP1A* are enriched in primary immunodeficiency as well (Figure 7B).

Furthermore, we constructed the PPI network to inspect the FKBP1A interactions with other proteins by using the STRING database. The PPI network showed that FKBP1A significantly interacted with 10 most related interactive proteins, which were ACVR1, RYR1, MTOR, TGFBR1, RYR2, PPP3CB, RPTOR, PPP3R1, PPP3CA, and SMAD7 (Figure 7C). In addition, STRING was used to perform KEGG analyses to determine the functional enrichment of these interactors. The results showed that regulation of these proteins was involved in some immune-related pathways such as Th17 cell differentiation (FDR = 6.40 × 10^−7^), PD-L1 expression and the PD-1 checkpoint pathway in cancer (FDR = 9.88 × 10^−6^), B-cell receptor signaling pathway (FDR = 0.00022), Th1 and Th2 cell differentiation (FDR = 0.00028), and T-cell receptor signaling pathway (FDR = 0.00037) (Table 3). Corresponding to our analysis above on LinkedOmics, the hepatocellular carcinoma pathway (FDR = 0.0255) was also found to be significantly associated with these protein networks (Table 3). Combined together, it ascertains that LIHC FKBP1A upregulation stays tightly association with the immunodeficiency pathway in LIHC patients but not in the LIHC patients infected with the hepatitis virus or consumers of alcohol.

### 2.7. Immune Infiltration Analysis of FKBP1A in LIHC

As immune cells play an essential role in angiogenesis and regulating immune escape in tumor progression, we investigated whether *FKBP1A* expression was correlated with immune infiltration levels in LIHC from TIMER database. We analyzed six main infiltrating immune cells (B cells, CD8^+^ T cells, CD4^+^ T cells, neutrophils, macrophages, and dendritic cells). Our data show that *FKBP1A* SCNAs (somatic copy number alterations) have significant correlations with infiltrating levels of B cells with arm-level deletion (Appendix A, *p* < 0.05). In addition, *FKBP1A* expression has a negative correlation with tumor purity (r = −0.318, *p* = 1.03 × 10^–2^) and significant positive correlations with infiltrating levels of B cells (r = 0.351, *p* = 2.20 × 10^−11^), CD8^+^ cells (r = 0.466, *p* = 8.11 × 10^−20^), CD4^+^ T cells (r = 0.279, *p* = 1.49 × 10^−7^), macrophages (r = 0.482, *p* = 3.24 × 10^−21^), neutrophils (r = 0.386, *p* = 1.05 × 10^−13^), and dendritic cells (r = 0.525, *p* = 1.84 × 10^−25^) (Figure 8A). Moreover, the higher expression of *FKBP1A* in macrophages was associated with a poor 5-year survival prognosis compared with the lower expression (Figure 8B, *p* < 0.05); and the higher expression was not associated with any significant differences in the 5-year survival rates in B cells, CD8^+^ T cells, CD4^+^ T cells, neutrophils, or dendritic cells of LIHC patients (Figure 8B, *p* > 0.05). Thus, our analysis proved that significant infiltration with macrophages appears to be a key factor in *FKBP1A* affecting the prognostic outcome of LIHC.

To inspect the prognostic values of *FKBP1A* expression and the abundance of different types of macrophage infiltration on LIHC patients’ survival, we, furthermore, generated Kaplan–Meier plots using the TIMER2.0 database with CIBERSORT algorithm. Remarkably, LIHC patients with low *FKBP1A* expression and low M2 macrophage infiltration had a better prognosis than those with low *FKBP1A* expression and high M2 macrophage infiltration (Figure 8C, HR = 1.63, *p* = 0.0456). For patients with high *FKBP1A* expression, high M2 macrophage infiltration indicated poorer survival than those with lower M2 macrophage infiltration (Figure 8C, HR = 1.6, *p* = 0.0282). 

As a further complication, our analysis also indicated that the infiltration of M0 or M1 macrophages correlating with the expression of *FKBP1A* had no significant effect on the prognosis of LIHC patients (Figure 8C, *p* > 0.05). The results specified that M2 macrophage infiltration, but not M0 or M1, may be one of the key reasons that caused *FKBP1A* to become a prognostic factor. Accordingly, we examined the expression correlation of *FKBP1A* with M2 macrophage markers in LIHC using TIMER2.0 database. The results showed that *FKBP1A* was significantly positively correlated with the expression of markers for M2 macrophages such as *CD68* (r = 0.382, *p* = 2.02 × 10^−13^), *CD163* (r = 0.168, *p* = 1.72 × 10^−3^) and *CD209* (r = 0.242, *p* = 5.42 × 10^−6^) (Figure 8D). These findings, above, strongly suggest that *FKBP1A* expression is positively linked to M2 macrophage infiltration in hepatocellular carcinoma.

As the ability of tumor cells to escape from tumor immunosurveillance contributes to cancer development [23] and tumors can evade immune responses by taking advantage of immune checkpoint genes [24]; therefore, we analyzed the genes of immune checkpoints on LIHC. Interestingly, our analysis identified a number of immune checkpoints including PD-1 (encoded by *PDCD1*), PD-L1 (encoded by *CD274*), CTLA-4, lymphocyte activation gene 3 protein (LAG3) and, T-cell immunoglobulin domain and mucin domain 3 (TIM3, also known as HAVCR2) [24]. 

We consequently analyzed *FKBP1A* and the expression of immune checkpoint genes in TIMER2.0 adjusted by tumor purity. *FKBP1A* expression was significantly and positively correlated with *PD-1* (r = 0.397, *p* = 1.9 × 10^−14^), *PD-L1* (r = 0.121, *p* = 2.46 × 10^−2^), *CTLA-4* (r = 0.473, *p* = 1.27 × 10^−20^), *LAG3* (r = 0.284, *p* = 7.67 × 10^−8^), and *HAVCR2* (r = 0.506, *p* = 7.06 × 10^−24^) (Appendix A). We carried out further analysis of the GEPIA database and the corresponding results were determined as the significant positive correlation of *FKBP1A* expression with *PD-1* (r = 0.28, *p* = 4.3 × 10^−8^), *PD-L1* (r = 0.55, *p* = 0), *CTLA-4* (r = 0.6, *p* = 0), *LAG3* (r = 0.32, *p* = 5.9 × 10^−10^), and *HAVCR2* (r = 0.36, *p* = 1.3 × 10^−12^) in LIHC (Appendix A). These results obtained from two databases demonstrate that tumor immune escape is involved in *FKBP1A*-mediated LIHC carcinogenesis.

## 3. Discussion

As LIHC is a highly malignant tumor with poor clinical prognosis and high mortality, it is imperative that advanced study of the LIHC oncogene be carried out to help to distinguish new and promising prognostic biomarkers and drug targets, which will improve the clinical efficacy against LIHC. In our present study, we found that the expression level of *FKBP1A* mRNA was significantly upregulated in LIHC, i.e., increasing in LIHC patients with high stage or grade. Immunohistochemical analysis of human liver tissue microarrays further confirmed that the expression of FKBP1A protein was significantly increased in LIHC tissues. The higher expression of FKBP1A protein accumulates mainly in endoplasmic reticulum along with significant upsurge in cytosol but not in nuclei. Additionally, the high expression of FKBP1A protein is associated with a poor prognosis in LIHC patients. The expression of the protein has a strong significantly positive correlation with the infiltration of various immune cells and immune checkpoints. These data strongly suggest that the upregulated expression of FKBP1A might promote tumor progression and may modulate tumor immunity by regulating the infiltration of immune cells. Thus, we claim that *FKBP1A* could be identified as a potential indicator of LIHC diagnosis.

Recent studies showed that the expression of *FKBP1A* in breast cancer was decreased significantly, and low expression of *FKBP1A* was associated with poor prognosis and increased resistance to chemotherapy [3,4]. Another study showed that *FKBP1A* was overexpressed in head and neck squamous cell carcinoma (HNSCC) and *FKBP1A* upregulation was strongly associated with lymph node metastasis and poor prognosis [19]. A report claimed that the expression of *FKBP1A* was elevated in LIHC and *FKBP1A* knockdown blocked cell proliferation, migration, and invasion and induced autophagy by regulating the PI3K/AKT/mTOR signaling pathway [25]. Our findings showed that the expression of *FKBP1A* was significantly elevated in LIHC patients. We also found that the high transcription level of *FKBP1A* was significantly associated with the overall survival and disease-specific survival of LIHC patients. A high level of *FKBP1A* was not conducive to the survival of LIHC patients, and may inhibit the patients’ response to other diseases to a certain extent. Furthermore, we found that a high level of *FKBP1A* accelerated LIHC exacerbation, by analyzing progression-free survival. Studies of relapse-free survival showed that *FKBP1A* expression was inversely associated with prognosis in LIHC patients treated with multiple therapies. Therefore, our findings suggest that it is feasible to use *FKBP1A* as a potential diagnostic and prognostic marker for hepatocellular carcinoma in clinical practice.

As FKBP1A was first identified in dendrite cells of peripheral blood, the convenient knockout mouse model of the *FKBP1A* gene was generated with homologous recombination. The mice with *FKBP1A* deletion experienced a more severe cardiac defect of septal development without ventricular septum accomplishment and underwent early postnatal lethality similar to ventricular septal defect (VSD) symptoms in humans. Because the septal tunnel was unclosed with extra massive growth of ventricular tubercles in the mouse along with the syndrome and cranial neural tube closure defects, the animals could not survive after birth [26]. The cardiac-specific overexpression of *FKBP1A* cDNA driven by MHC promoter was carried out in mice as well. The cardiac phenotype caused by the gain of the *FKBP1A* gene experienced critical sudden cardiac death with high-grade conduction system dysfunction leading to potassium channel disorder [27]. To date, the molecular mechanism of how *FKBP1A* functions in liver hepatocellular carcinoma remains unidentified and our data provide valuable clues as to how the FKBP1A-RyR complex may be critically involved in pathological LIHC.

To date, increasing investigation of the tumor immune microenvironment has transformed the precision of the medical treatment of cancer. Regarding LIHC development associated with an inflammatory environment, the FKBP1A protein could be involved in the immune microenvironment of *LIHC*, based on bioinformatics analysis. The expression level of *FKBP1A* was significantly positively correlated with the infiltration of various immune cells (B cells, CD4^+^ T cells, CD8^+^ T cells, macrophages, neutrophils, and dendritic cells). The patients with higher M2 macrophage infiltration indicated an unfavorable prognosis; *FKBP1A* expression was markedly positively correlated with biomarkers of the M2 macrophages. These findings indicated that tumor immune cell infiltration, in particular M2 macrophages, might partially account for *FKBP1A*-mediated oncogenic roles in LIHC. The M2-polarized macrophages could be tumor-associated macrophages (TAMs). TAMs could be a major component of the tumor microenvironment. The TAMs are associated with tumor growth, metastasis and progression via the induction of extracellular matrix remodeling, angiogenesis, and therapeutic resistance [28]. Increasing studies have shown that the infiltration of M2 macrophages in the microenvironment of liver tumor indicates poor prognosis. The M2 macrophages promote liver tumor growth, epithelial-mesenchymal transition, migration and invasiveness [29,30,31]. Therefore, we claim that high *FKBP1A* expression resulting in poor prognosis may regulate tumor immune cell infiltration into the immune microenvironment of LIHC, especially represented by M2 macrophages. According to our data, there is increased expression of *FKBP1A* in dendrite cells and M2 macrophages of peripheral blood in LIHC. It could be possible that collecting and evaluating the *FKBP1A* expression in patients’ peripheral blood may be a practical approach for clinical therapeutic diagnostics.

Immune tolerance plays an important role in the development of LIHC and Iñarrairaegui et al. recently suggested that immune-checkpoint inhibition can be employed as an effective target in the therapeutic strategy [32]. Hepatocellular carcinoma and others can evade antitumor immune response by exploiting this physiological mechanism by expressing the corresponding ligands including *PD-L1* in tumor and stromal cells [5,33]. Regular treatments for LIHC include surgery, radiation therapy and targeted therapies are based on tyrosine protein kinase inhibitors [17,34]. However, the prognosis of hepatocellular carcinoma is very poor due to drug resistance and frequent recurrence and metastasis. Recently, new therapeutic strategies such as immunosuppressive therapy for cancer depending on ICIs have shown very promising results [17,35,36,37]. The combination of conventional therapies and immunotherapy can achieve greater efficacy through further synergistic effects in LIHC [17,37]. In our study, the high level of *FKBP1A* expression was significantly positively correlated with immune checkpoints including *PD-1*, *PD-L1*, *CTLA-4*, *LAG3* and *HAVCR2* in LIHC. Our results provide extra information that immunotherapy targeting the *FKBP1A* protein might increase the efficacy for LIHC treatment. Therefore, *FKBP1A* may serve as a potential target to increase the effectiveness of immunotherapy in LIHC. 

Our study here shows that the high expression of *FKBP1A* in LIHC patients may regulate the TGF-β signaling pathways to speed up the multiplication including cell proliferation, apoptosis or autophagy, inhibition of angiogenesis and inflammatory signals of the cancer cells according to KEGG pathway analysis from the STRING database. It was reported that TGF-β family polypeptides regulate a wide range of biological processes including growth and differentiation [38]. A previous report showed that FKBP12 was a physiological regulator of the cell cycle [39]. The knockout of FKBP12 in fibroblasts led to these cells growing much more slowly than wild-type cells. FKBP12-deficient cells manifest cell cycle arrest in the G1 phase, which was accompanied by a significant increase in p21 (WAF1/CIP1) expression [39]. The p21 upregulation-mediated cell cycle arrest was due to the over-activation of transforming growth factor-β (TGF-β) receptor signaling, which was inhibited by FKBP12 in wild type cells [39]. A further study considered that FKBP12 interacting with TGF-β type I receptors acted as a negative regulator of TGF-β receptor endocytosis [40]. Interestingly, Chen et al. reported that LIHC patients with an inactivation of TGF-β signaling experienced a lower survival time than patients with a normal or activated TGF-β signaling group [41]. LIHC patients with inactivation of TGF-β signature showed a loss of tumor suppressive function and decreased DNA repair activity [41,42]. Our results showing the high expression of *FKBP1A* regulating TGF-β signaling pathways in LIHC patients and accelerating multiplication, such as cell proliferation and immune cell infiltration, support the studies we described above.

Based on online databases and confirmed in the human liver samples chips, our study elucidated that *FKBP1A* was highly expressed in LIHC patients and positively correlated with poor prognosis. The *FKBP1A* expression in LIHC characterizes a strong potential as a novel prognostic and diagnostic biomarker for LIHC carcinogenesis. Our study provides recent evidence that *FKBP1A* is a potential prognostic factor in LIHC, since *FKBP1A* may be involved in immune cell infiltration-related signaling pathways to mediate LIHC development. 

The development and progression of hepatocellular carcinoma involves alterations in multiple signaling pathways. Each patient may respond differently to treatment modalities such as chemotherapy, radiation therapy, and immunotherapy. According to the cBioPortal databases, a few mutants occurred in exons encoding *FKBP1A*, of which not all variations were statistics related to LIHC. It seems the information embedded in the genomic introns should be investigated intensively for a more comprehensive study. A current study linking LncRNA AFAP1-AS1 modulation and sensitivity to paclitaxel via the miR-195-5p/FKBP1A axis [20] may give us clues to inspect the mutation in the genomic non-coding zone and its fundamental function in LIHC. Precision therapeutic approaches to cancer treatment, known as precision oncology, use the molecular characteristics of an individual patient’s tumor to assess the likelihood of benefit or toxicity of a specific therapeutic intervention [43]. It is imperative to search for specific tumor molecular markers, select sensitive populations, and, fundamentally, realize tailor-made individualized treatment. In the future, different basic experiments and large-scale clinical trials are needed to explore the role of *FKBP1A* in LIHC. More precision and individualized approaches need to be tested in well-designed clinical trials. It will be important to validate the clinical significance of *FKBP1A* and to investigate the potentiality of *FKBP1A* as a prognostic biomarker in the future.

## 4. Materials and Methods

### 4.1. Oncomine Database Analysis

The Oncomine database is the world’s largest oncogene chip database and integrated data-mining tool (https://www.oncomine.org/resource/login.html, accessed on 22 September 2021). The Oncomine platform is publicly accessible online and collects the most complete spectra of cancer mutations, related gene expression profiles, and relevant clinical information [44]. To examine the mRNA expression of *FKBP1A* between tumors and normal tissues in different types of carcinomas, a Student’s *t*-test was performed on this data with the significance threshold set as follows: Gene = *FKBP1A*; *p*-value = 0.05; fold change = 2; gene rank: 10%; data type: mRNA.

### 4.2. TIMER Database Analysis

The Tumor Immune Estimation Resource (TIMER, https://cistrome.shinyapps.io/timer/, accessed on 8 May 2022) is a public resource for systematic analysis of immune infiltration in samples from The Cancer Genome Atlas (TCGA) [45,46,47]. The expression of *FKBP1A* was determined under the Gene DE module with default parameters in TIMER2.0. The correlation of *FKBP1A* expression with the abundance of six tumor-immune infiltration cells (B cells, CD4^+^ T cells, CD8^+^ T cells, neutrophils, macrophages, and dendritic cells) in LIHC was analyzed by the TIMER algorithm database. In addition, under the ‘‘Outcome Module’’ in TIMER2.0, we explored the clinical relevance of tumor immune subsets of macrophages [32]. The generated scatter plots suggest statistical significance and provide the purity-corrected partial Spearman’s rho value. Gene expression values were transformed to log2 PTM values. *p* < 0.05 was statistically significant.

### 4.3. GEPIA Database Analysis

The Gene Expression Profiling Interactive Analysis resource (GEPIA, http://gepia.cancer-pku.cn/index.html, accessed on 8 May 2022) was used to compare the *FKBP1A* expression levels in normal liver and tumor tissues based on the Genotype-Tissue Expression (GTEx) and TCGA databases with the thresholds of |Log2 (Fold Change) | Cut-off: 1 and *p*-value Cut-off: 0.01 [48]. We also used GEPIA to generate survival curves for DFS (disease-free survival) and OS (overall survival) by classifying the patients into high and low *FKBP1A* expression groups based on the median *FKBP1A* expression value. The *p* value was calculated using the logrank test and the hazards ratio (HR) and 95% confidence intervals (CI) were also calculated. The log-rank test *p* < 0.05 indicated the significance of survival time. GEPIA was also used to analyze the *FKBP1A* expression correlation with *PD-1*, *PD-L1*, *CTLA-4*, *LAG3* and *HAVCR2* in LIHC using Spearman’s correlation analysis.

### 4.4. HCCDB Database Analysis

The Hepatocellular Carcinoma Database (HCCDB, http://lifeome.net/database/hccdb/home.html, accessed on 8 May 2022) is an integrated liver cancer database containing 15 public hepatocellular carcinoma gene expression datasets including the data from the Gene Expression Omnibus (GEO), Liver Hepatocellular Carcinoma Project of The Cancer Genome Atlas (TCGA-LIHC) and Liver Cancer-RIKEN, and the JP Project from International Cancer Genome Consortium (ICGC LIRI-JP) [49]. HCCDB was used to analyze the *FKBP1A* gene expression level between tumor samples and adjacent samples in each dataset.

### 4.5. Kaplan–Meier Plotter Analysis

The Kaplan–Meier plotter (http://kmplot.com/analysis/, accessed on 8 May 2022) could evaluate the effect of more than 54,000 biomolecules on the survival rate in more than 10,000 cancer samples by utilizing the RNA-seq data in the TCGA, EGA, and GEO databases [50,51]. The Kaplan–Meier plotter was used to evaluate the influence of *FKBP1A* on prognostic value including disease-specific survival (DSS), relapse-free survival (RFS), progression-free survival (PFS), and overall survival (OS). We also analyzed the prognostic value (OS and PFS) of *FKBP1A* in LIHC patients with diverse clinicopathologic features such as cancer stage, grade, AJCC-T, gender, vascular invasion, race, alcohol consumption and viral hepatitis. The best cut-off value was determined from all possible cut-off values between the lower and the upper quartiles. The relative prognostic value of *FKBP1A* on routine clinicopathological features in the TCGA was evaluated with univariate/multivariate Cox proportional hazards analysis. The hazard ratio (HR) with 95% confidence intervals and log-rank *p*-value were also computed. *p* < 0.05 was statistically significant.

### 4.6. LinkedOmics Database Analysis

The LinkedOmics database (http://www.linkedomics.org/login.php, accessed on 8 May 2022) is a publicly available portal server for analyzing multidimensional datasets based on TCGA [52]. The LinkFinder module of LinkedOmics was used to study the differentially expressed genes related to *FKBP1A* in the TCGA LIHC cohort. The correlation of results was tested by the Pearson correlation coefficient and presented, respectively, in volcano plot and heat maps. Gene Ontology (GO) (CC (cellular component), BP (biological process), and MF (molecular function)) and KEGG (Kyoto Encyclopedia of Genes and Genomes) pathways were performed by the gene set enrichment analysis (GSEA) in the LinkInterpreter module [53]. The rank criterion was FDR (false discovery rate) < 0.05, a minimum number of genes of 3, and a simulation of 500 in the LIHC dataset.

### 4.7. UALCAN Database Analysis

UALCAN (http://ualcan.path.uab.edu/index.html, accessed on 8 May 2022) is an open interactive web-portal for the performance of in-depth analyses of TCGA gene expression data using TCGA level 3 RNA-seq and clinical data from 31 cancer types [54]. UALCAN was utilized to analyze the mRNA expression of *FKB1A* in LIHC patients based on individual cancer stages, tumor grade and nodal metastasis status. The *p* value cutoff was 0.05.

### 4.8. STRING Database Analysis

STRING database (https://string-db.org/, accessed on 8 May 2022) covers more than 5000 organisms and 24.6 million proteins. In addition to the experimental data, the results of text mining from PubMed abstracts and other database data, it also contains the results predicted by bioinformatics methods [55,56]. It can be used for searching known and predicted protein–protein interactions and performing gene-set enrichment analysis. Here, we used STRING to analysis the potential protein–protein interaction network with FKBP1A and KEGG gene-set enrichment.

### 4.9. Tissue Microarrays (TMA) and Immunohistochemistry (IHC)

#### 4.9.1. Information on Human Liver Tissue Microarray

A commercial human liver tissue microarray, comprising 79 LIHC patients, 10 adjacent-to-tumor samples and 6 healthy individuals was purchased from the Bioaitech (cat. # D950601; Bioaitech Co., Ltd., Xi’an, China). These tissues were formalin fixed and then paraffin embedded. The tissue chips were provided in 10 μm thickness of microtone sections. All experiments with human samples for immunohistochemical and pathological analysis described in this study were covered by the Medical Ethics Committee of Shaanxi Normal University (Xi’an City, Shaanxi Province, China) and were carried out in accordance with the Helsinki Declaration

#### 4.9.2. The Process of Immunohistochemical and Immunofluorescence Staining

The TMA slide was baked in the oven at 60 °C for 0.5 h followed by conventional deparaffinization and rehydration. After that, antigen retrieval was carried out with sodium citrate buffer (pH 6.0). Endogenous peroxidase was blocked with 3% H_2_O_2_-methanol for 10 min and washed with phosphate-buffered saline (PBS) for 5 min × 3 times. The slide then was blocked with 3% bovine serum albumin (BSA) in PBS for 1 h at room temperature, and then further incubated with FKBP1A antibody (cat. no. sc-28814; 1:100; Santa Cruz Biotechnology, Dallas, TX, USA) overnight at 4 °C. After washing with PBS containing 0.05% Tween-20, the TMA slide was subsequently treated with peroxidase-labelled polymer conjugated to goat anti-rabbit immunoglobulins (Dako EnVision HRP; Dako, Copenhagen, Denmark) for 60 min and visualized with diaminobenzidine after an incubation for 2 min at room temperature. Finally, the slide was counterstained with hematoxylin, dehydrated, and covered. For immunofluorescent staining of anti-RyR (1:100; Cat No. sc-34019, Santa Cruz Biotechnology, Dallas, TX, USA) and anti-FKBP1A (cat. no. sc-28814; 1:100; Santa Cruz Biotechnology, Dallas, TX, USA) antibody, the slides were incubated with the antibody at 4 °C overnight, followed by incubation for 1 h with Fluorescein (FITC) AffiniPure Donkey Anti-Goat IgG (H+L) (1:100, Cat No. 705-095-003 Jackson ImmunoResearch Inc., West Grove, PA, USA) and Rhodamine Red-X (RRX) AffiniPure Goat Anti-Rabbit IgG (H+L) (1:100, Cat No. 111-295-003 Jackson ImmunoResearch Inc., West Grove, PA, USA) at room temperature. DAPI was used to visualize nuclei.

#### 4.9.3. Statistical Analysis of IHC

Representative images from each sample were collected using a 20 × objective lens and quantitative assessment was made of IHC images of human tissue samples by IHC Profiler plugin of image J [57]. Statistical significance was determined with two-tailed Student’s t-test between two groups using GraphPad Prism 7 software (GraphPad Software Inc., San Diego, CA, USA). A value of *p* < 0.05 was considered statistically significant.

## 5. Conclusions

The expression of *FKBP1A* is significantly elevated in patients with LIHC, and a high expression of *FKBP1A* is correlated with poor prognosis. Interestingly, the expression level of FKBP1A protein is also significantly elevated in LIHC patients and accumulates in the ER of hepatocytes. This suggests that ER accumulation of FKBP1A plays an important role in promoting the development of LIHC. Its high expression in LIHC patients promotes immune cell infiltration in the tumor microenvironment characterized by B cells, CD8^+^ T cells, CD4^+^ T cells, macrophages, neutrophils, and dendritic cells with M2 macrophage significant association for a poor survival prognosis. The increased *FKBP1A* triggers the expression of immune checkpoint genes such as *PD-L1*, *CTLA-4*, *LAG3* and *HAVCR2*. These results demonstrate that *FKBP1A* could be a potential diagnostic and prognostic target associated with immune infiltration in LIHC.

## Figures and Tables

**Figure 1 ijms-23-12797-f001:**
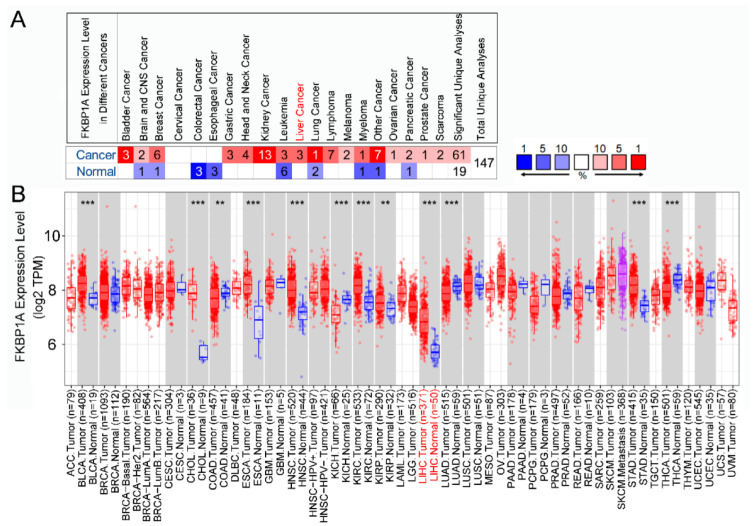
The *FKBP1A* expression levels in various types of cancer tissues indicated by mRNA detection. (**A**) The expression level of *FKBP1A* in different types of tumor tissues and normal tissues in the Oncomine database. Number listed under name of each tumor is sample number (*n*) recorded in each dataset. Red highlight indicates high expression and blue indicates low expression. Darker red or darker blue represents higher or lower expression, respectively. (**B**) The expression levels of *FKBP1A* in different types of tumor tissues queried from the TCGA Oncomine tumor database and normal tissues via TIMER2.0 database analysis. ** *p* < 0.01, *** *p* < 0.001.

**Figure 2 ijms-23-12797-f002:**
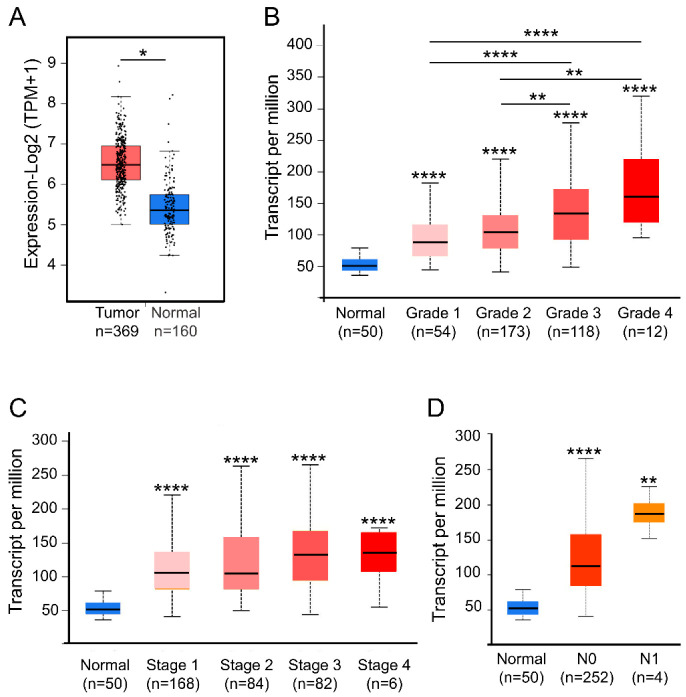
*FKBP1A* transcription level was significantly increased in LIHC patients. (**A**) Box plots of *FKBP1A* mRNA expression in normal liver and tumor samples based on GEPIA database. (**B**–**D**) *FKBP1A* mRNA level in LIHC patients based on different variables by using the UALCAN database. (**B**) Normal individuals or LIHC patients with grade 1, 2, 3, or 4 tumors. (**C**) Normal individuals or in LIHC patients in stage 1, 2, 3, or 4. (**D**) Normal individuals and LIHC patients with lymph node metastasis. The unpaired Student’s *t* test was used to estimate the significance of difference in gene expression levels between groups. * *p* < 0.05, ** *p* < 0.01, **** *p* < 0.0001 vs. normal.

**Figure 3 ijms-23-12797-f003:**
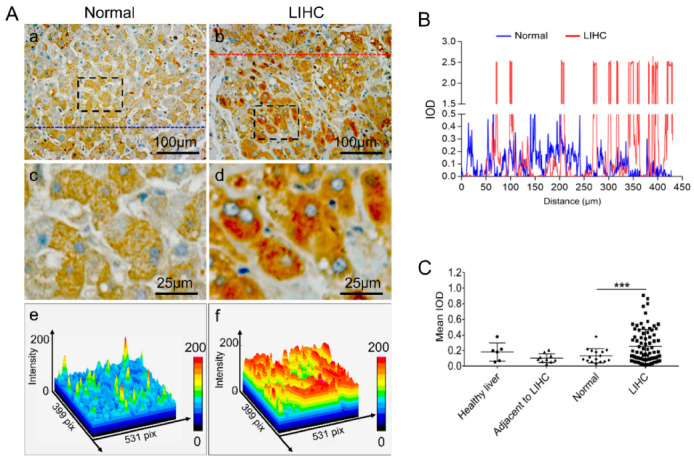
Human tissue microarray including hepatocellular carcinoma and normal liver tissue analysis revealed that the expression of FKBP1A was significantly elevated in LIHC patients. (**A**) Representative IHC images of FKBP1A in normal liver (**a**) or LIHC patients (**b**). (**c**) Shows the enlarged image of the black box in (**a**); (**d**) is the enlarged image of the black box in (**b**). The density distribution of (**c**,**d**) are shown in panels (**e**,**f**). n (normal) = 16; n (LIHC) = 79. (**B**) The plot profiles of the intensity of FKBP1A protein in hepatocytes of normal tissues and LIHC along the straight line (as shown (**a)** and (**b**). (**C**) Quantification of the mean optical density (mean IOD) of FKBP1A protein by IHC staining in healthy liver, adjacent to LIHC or LIHC tissues. The normal group includes healthy liver and adjacent to LIHC tissues. n (healthy liver) = 6; n (adjacent to LIHC) = 10; n (normal) = 16; n (LIHC) = 79. Data are presented as the means ± SD. *** *p* < 0.001.

**Figure 4 ijms-23-12797-f004:**
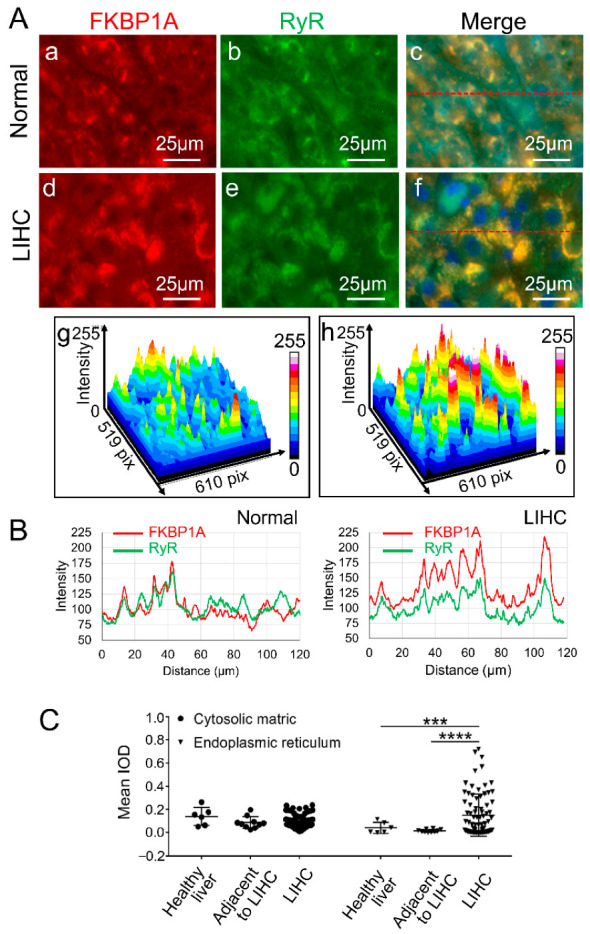
FKBP1A expression distribution is accompanied with ER accumulation in LIHC based on human liver microarray analysis. (**A**) Representative images of immunofluorescence staining of endoplasmic reticulum marker RyR (green), FKBP1A (red) and nucleus (blue) from normal (**a**–**c**) and LIHC (**d**–**f**). The co-localization regions of FKBP1A and RyR are shown in yellow (**c**,**f**). (**g**,**h**) The plot profiles of the intensity of co-localization regions of FKBP1A and RyR in hepatocytes of normal tissues and LIHC along the straight line (as shown **c**,**f**). n (normal) = 16; n (LIHC) = 79. (**B**) The plot profiles of the intensity of FKBP1A and RyR in hepatocytes of normal tissues and LIHC along the straight line (as shown **c**,**f**). (**C**) Quantification of the mean optical density (mean IOD) of FKBP1A protein in healthy liver, adjacent to LIHC or LIHC tissues. The normal group includes healthy liver and adjacent to LIHC tissues. n (healthy liver) = 6; n (adjacent to LIHC) = 10; n (normal) = 16; n (LIHC) = 79. Data are presented as the means ± SD. *** *p* < 0.001, **** *p* < 0.0001.

**Figure 5 ijms-23-12797-f005:**
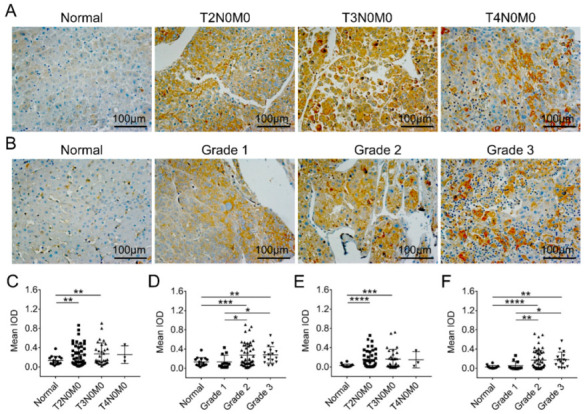
Human liver tissue microarray analysis for the correlation between FKBP1A protein expression level and LIHC patients depending upon TNM staging or histologic grade. (**A**,**B**) Representative images of human normal liver tissues and LIHC patients with T2N0M0, T3N0M0, or T4N0M0 tumors (**A**) and grade 1, 2, or 3 tumors (**B**) with FKBP1A staining. (**C**,**D**) Quantification of the mean optical density (mean IOD) of FKBP1A protein at different TNM stages (C) and different tumor grades (**D**). (**E**,**F**) Quantification of the mean IOD of FKBP1A protein distributed in the endoplasmic reticulum at different TNM stages (**E**) and in different tumor grades (**F**). The normal group includes healthy liver tissues and adjacent to LIHC tissues. Tumor grade is the description of a tumor based on how abnormal the tumor cells and the tumor tissue look under a microscope. Each point represents an individual tissue. Scale bar, 100 μm. Data are presented as the means ± SD. * *p* < 0.05, ** *p* < 0.01, *** *p* < 0.001, **** *p* < 0.0001.

**Figure 6 ijms-23-12797-f006:**
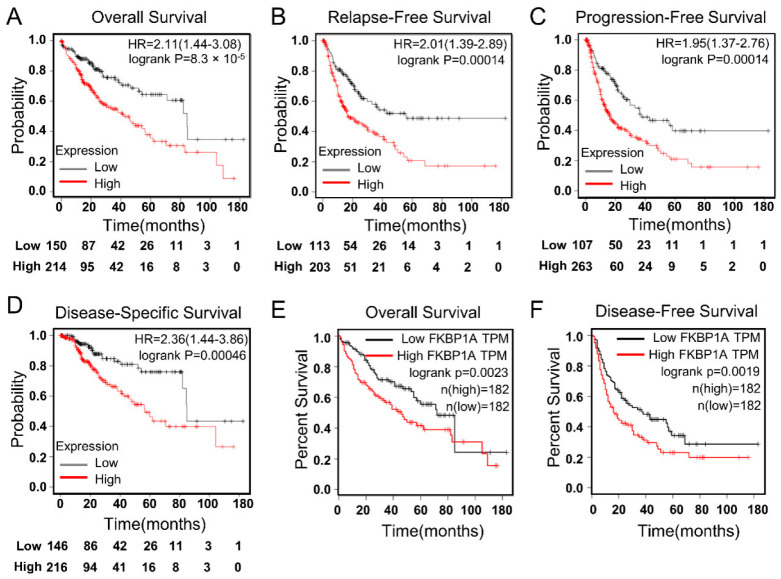
High *FKBP1A* expression predicts poor prognosis in LIHC patients. (**A**–**D**), Correlation analysis between *FKBP1A* expression and prognostic survival in LIHC patients via Kaplan–Meier plotter analysis stratified at “best cut-off”. The hazard ratio (HR) was calculated based on the Cox proportional hazards model. (**A**) Overall survival, *n* = 364; (**B**) relapse-free survival, *n* = 316; (**C**) progression-free survival, *n* = 370; (**D**) disease-specific survival, *n* = 362. (**E**,**F**) Overall survival (**E**) and disease-free survival (**F**) curves of *FKBP1A* in LIHC patients assessed in the GEPIA database. The median value was used as the cutoff for dividing the high and low groups. The log-rank method was used for the hypothesis test.

**Figure 7 ijms-23-12797-f007:**
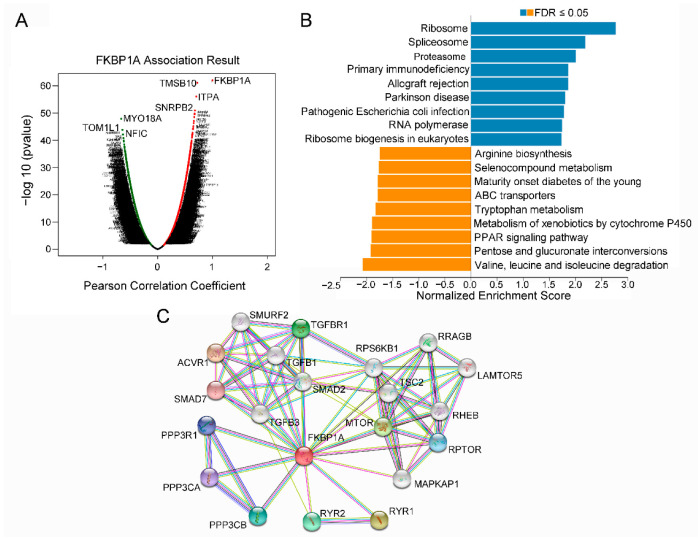
The co-expression genes with *FKBP1A* and function of enrichment analysis concerning *FKBP1A*-related genes in LIHC. (**A**) All the genes significantly associated with *FKBP1A* distinguished by Pearson test in LIHC cohort (LinkedOmics database). Red and green dots represent positively and negatively significantly correlated genes with *FKBP1A*, respectively. (**B**) KEGG pathway analysis. Dark blue and orange indicate FDR ≤ 0.05. FDR, false discovery rate. (**C**) A protein–protein interaction (PPI) network of FKBP1A protein from STRING database.

**Figure 8 ijms-23-12797-f008:**
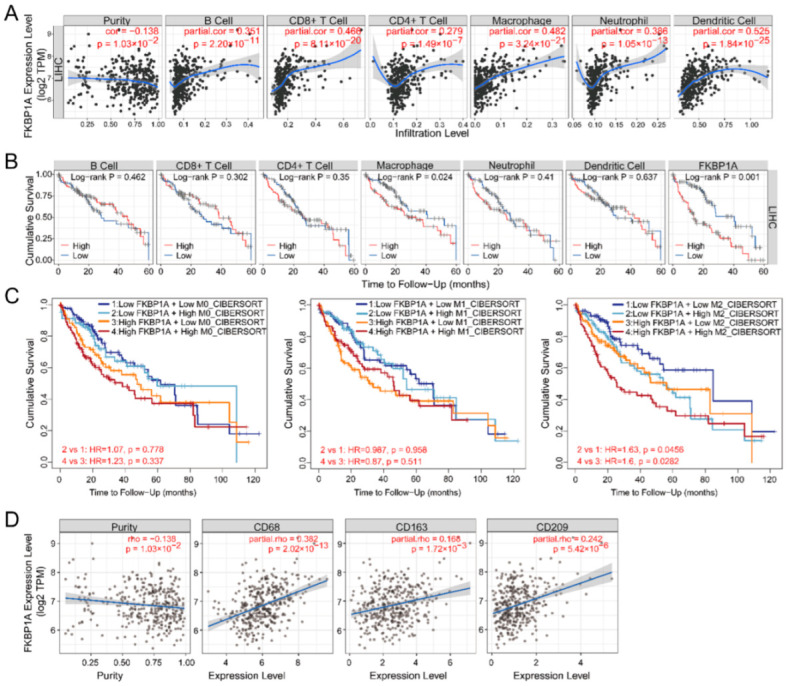
Correlation analysis between *FKBP1A* expression and tumor-infiltrating immune cells in LIHC. (**A**) *FKBP1A* expression is significantly related to tumor purity and has significant positive correlations with infiltrating levels of B cells, CD8^+^ T cells, CD4^+^ T cells, macrophages, neutrophils, and dendritic cells obtained from TIMER (purity-corrected Spearman test) in LIHC. (**B**) Five-year overall survival curve of the six tumor-infiltrating immune cells and *FKBP1A* expression in LIHC patients produced by Kaplan–Meier estimator from TIMER. Survival differences are compared between patients with high and low (split percentage of patients is 30%) infiltration of each kind of immune cells. Log-rank *p* < 0.05 is considered significant. (**C**) The Kaplan–Meier plot from the TIMER2.0 shows the difference of overall survival among patients stratified by both the estimated infiltration level of different types of macrophages (M0, M1 and M2) and *FKBP1A* expression level in LIHC. (**D**) Scatter plots from the “Gene Module”. Correlation of *FKBP1A* expression with tumor purity and with the infiltration level of M2 macrophage markers *CD68*, *CD163* and *CD209*. *p* < 0.05 is considered significant.

**Table 1 ijms-23-12797-t001:** Correlation analysis of *FKBP1A* expression and prognostic overall survival in LIHC patients with different clinic-pathological factors by using Kaplan–Meier plotter.

Clinicopathological Factors		Overall Survival (*n* = 364)			Progression-Free Survival (*n* = 366)		
		N	Hazard Ratio	*p* Value	N	Hazard Ratio	*p* Value
Sex	Female	118	1.82 (1.04–3.18)	**0.033**	121	1.87 (1.08–3.22)	**0.022**
	Male	246	2.55 (1.5–4.32)	**0.00034**	249	2.2 (1.4–3.45)	**0.00048**
Stage	1	170	2.54 (1.38–4.66)	**0.0019**	171	2.03 (1.13–3.65)	**0.016**
	2	83	0.62 (0.25–1.54)	0.2978	85	1.88 (0.94–3.73)	0.0678
	3	83	3.52 (1.55–7.97)	**0.0014**	85	1.98 (1.14–3.44)	**0.0142**
	4	4	-	-	5	-	-
Grade	1	55	3.8 (1.41–10.25)	**0.0048**	55	2.69 (1.23–5.88)	**0.0099**
	2	174	1.79 (1.07–2.99)	**0.0251**	177	1.84 (1.14–2.95)	**0.0107**
	3	118	2.18 (1.18–4.02)	**0.0106**	121	1.55 (0.84–2.85)	0.1612
	4	12	-	-	12	-	-
AJCC-T	1	180	2.34 (1.31–4.19)	**0.0032**	181	1.94 (1.11–3.38)	**0.0175**
	2	90	0.61 (0.27–1.37)	0.2278	93	1.93 (0.98–3.78)	0.0519
	3	78	2.79 (1.45–5.37)	**0.0014**	80	2.2 (1.12–4.29)	**0.0187**
	4	13	-	-	13	-	-
Vascular invasion	None	203	1.88 (1.1–3.19)	**0.0182**	205	1.67 (1.04–2.67)	**0.0322**
	Micro	90	2.27 (0.78–6.62)	0.1211	92	2.5 (1.24–5.04)	**0.0079**
Race	White	181	1.76 (1.1–2.8)	**0.0163**	184	2.22 (1.37–3.59)	**0.0008**
	Asian	155	4.12 (1.74–9.75)	**0.0005**	157	2.72 (1.45–5.08)	**0.0012**
Alcohol consumption	Yes	115	2.31 (1.13–4.74)	**0.0191**	116	1.55 (0.91–2.67)	0.1059
	None	202	2.09 (1.31–3.32)	**0.0016**	205	2.83 (1.65–4.87)	**8.8 × 10^−5^**
Viral hepatitis	Yes	160	2.27 (1.07–4.82)	**0.0276**	153	1.63 (0.93–2.84)	0.086
	None	167	1.92 (1.19–3.08)	**0.0063**	169	2.56 (1.51–4.34)	**0.0003**

Bold values indicate *p* < 0.05.

**Table 2 ijms-23-12797-t002:** Prognostic factors for overall survival in univariable and multivariable Cox proportional hazards analyses.

Factor	Univariable Analysis		Multivariable Analysis	
	Hazard Ratio (95% CI)	*p*-Value	Hazard Ratio (95% CI)	*p*-Value
*FKBP1A*	1.766 (1.375–2.268)	<0.001	1.546 (1.185–2.017)	0.001
Age	1.012 (0.999–1.026)	0.078	1.008 (0.994–1.022)	0.261
Gender	0.816 (0.573–1.163)	0.260	0.907 (0.623–1.320)	0.611
pT stage	1.675 (1.397–2.007)	<0.001	1.804 (1.411–2.306)	<0.001
pTNM stage	1.376 (1.145–1.654)	<0.001	0.870 (0.687–1.101)	0.246
Grade	1.121 (0.887–1.417)	0.339	1.079 (0.836–1.392)	0.559

**Table 3 ijms-23-12797-t003:** LIHC-associated KEGG pathway analysis based on STRING database.

#Term ID	Term Description	Count in Network	Strength	False Discovery Rate
hsa04659	Th17 cell differentiation	5/101	1.94	6.40 × 10^−7^
hsa04218	Cellular senescence	5/150	1.77	2.11 × 10^−6^
hsa04921	Oxytocin signaling pathway	5/149	1.78	2.11 × 10^−6^
hsa04020	Calcium signaling pathway	5/193	1.66	3.72 × 10^−6^
hsa05235	PD-L1 expression and PD-1 checkpoint pathway in cancer	4/88	1.91	9.88 × 10^−6^
hsa05020	Prion disease	5/265	1.53	1.16 × 10^−5^
hsa04380	Osteoclast differentiation	4/122	1.77	2.50 × 10^−5^
hsa04371	Apelin signaling pathway	4/131	1.73	2.89 × 10^−5^
hsa05167	Kaposi sarcoma-associated herpesvirus infection	4/187	1.58	0.0001
hsa04370	VEGF signaling pathway	3/57	1.97	0.00013
hsa05163	Human cytomegalovirus infection	4/218	1.51	0.00013
hsa05166	Human T-cell leukemia virus 1 infection	4/211	1.53	0.00013
hsa05170	Human immunodeficiency virus 1 infection	4/204	1.54	0.00013
hsa04720	Long-term potentiation	3/64	1.92	0.00015
hsa04924	Renin secretion	3/66	1.91	0.00015
hsa05031	Amphetamine addiction	3/66	1.91	0.00015
hsa04662	B cell receptor signaling pathway	3/78	1.84	0.00022
hsa04010	MAPK signaling pathway	4/288	1.39	0.00028
hsa04658	Th1 and Th2 cell differentiation	3/87	1.79	0.00028
hsa04350	TGF-beta signaling pathway	3/91	1.77	0.00029
hsa04625	C-type lectin receptor signaling pathway	3/102	1.72	0.00037
hsa04660	T-cell receptor signaling pathway	3/101	1.72	0.00037
hsa04922	Glucagon signaling pathway	3/101	1.72	0.00037
hsa04724	Glutamatergic synapse	3/111	1.68	0.00043
hsa05010	Alzheimer disease	4/355	1.3	0.00043
hsa05014	Amyotrophic lateral sclerosis	4/352	1.31	0.00043
hsa04114	Oocyte meiosis	3/120	1.65	0.00048
hsa04650	Natural killer cell mediated cytotoxicity	3/121	1.64	0.00048
hsa04310	Wnt signaling pathway	3/154	1.54	0.00092
hsa04022	cGMP-PKG signaling pathway	3/162	1.52	0.001
hsa05152	Tuberculosis	3/168	1.5	0.0011
hsa04136	Autophagy—other	2/29	2.09	0.0013
hsa04360	Axon guidance	3/177	1.48	0.0013
hsa04213	Longevity regulating pathway—multiple species	2/61	1.77	0.0054
hsa05212	Pancreatic cancer	2/73	1.69	0.0075
hsa05210	Colorectal cancer	2/82	1.64	0.0091
hsa04211	Longevity regulating pathway	2/87	1.61	0.0099
hsa04713	Circadian entrainment	2/92	1.59	0.0108
hsa04152	AMPK signaling pathway	2/120	1.47	0.0176
hsa04728	Dopaminergic synapse	2/128	1.44	0.0194
hsa04140	Autophagy—animal	2/130	1.44	0.0195
hsa04910	Insulin signaling pathway	2/133	1.43	0.0199
hsa05017	Spinocerebellar ataxia	2/135	1.42	0.02
hsa05226	Gastric cancer	2/144	1.39	0.0222
hsa04150	mTOR signaling pathway	2/151	1.37	0.0238
hsa04390	Hippo signaling pathway	2/153	1.37	0.0239
hsa05206	MicroRNAs in cancer	2/160	1.35	0.0255
hsa05225	Hepatocellular carcinoma	2/160	1.35	0.0255
hsa05131	Shigellosis	2/218	1.21	0.0443
hsa04714	Thermogenesis	2/229	1.19	0.0478

## Data Availability

Publicly available datasets were analyzed in this study as described in Section 4. Each portal’s persistent web link has been supplied.

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
