# Peer review of "The Prognostic Significance of FKBP1A and Its Related Immune Infiltration in Liver Hepatocellular Carcinoma"

_ijms, 2022, doi:10.3390/ijms232112797_

Round 1
Reviewer 1 Report
NONE
Author Response
Author response to reviewer comment: We thank the reviewer for taking efforts and spending time to review our manuscript.
Reviewer 2 Report
In this manuscript Li et al. have investigated the expression levels of FKBP1A and their diagnostic and prognostic value in liver hepatocellular carcinoma (LIHC). Based on the information provided by multiple on-line data bases, and some results from immunohistochemical analyses of commercially available human liver sample chips, the authors found that the expression of FKBP1A increased in the liver of LIHC patients, mainly localizing on the endoplasmic reticulum of hepatocytes. Moreover, they found that high expression of FKBP1A correlated with malignant tumor growth and poor survival of LIHC patients, being associated with infiltration of immune cells and expression of specific markers of M2 macrophages and immune checkpoint receptors, suggesting that FKBP1A could be a potential prognostic target involved in LIHC immune cell infiltration.
Major comment
The work by Li et al. represents an interesting approach to analyse the relationship between the immunophilin FKBP1A and liver hepatocellular carcinoma. Helped by detailed information from public databases, the authors have found a potential use of FKBP1A expression as prognosis factor to evaluate disease progression in LIHC patients. From this point of view, the work may be considered new and interesting. However, results are merely correlative and no insights about the possible role played by FKBP1 in LIHC are provided. In this regard, additional important questions remain. For example, whether FKBP1A is certainly involved in the mechanism of tumorigenesis. Additional experimental work should be thus required to demonstrate the authors’ suggestion that FKBP1A is involved in tumor immune cell infiltration in LIHC. Any evidence on this direction will considerably increase the manuscript’s interest and novelty.
Some specific points
1.- Figure 3A-C: The number of microscopy fields that have been analysed and the number of controls and patients that have been studied should be indicated on the figure legend.
2.- Figure 3B and 3C: Why is the background expression of FKBP1A (mean optical densities) so different between normal and LIHC immunohistochemistry? Specific values of FKBP1A expression in normal and LIHC samples should be normalized to their respective backgrounds.
3.- Figure 3D and E: The number of samples, donors and patients analysed in immunohistochemistry studies should be also indicated. On the other hand, immunofluorescence confocal microscopy analysis of co-localization between FKBP1A and an ER specific marker is required to characterize in depth the cellular localization of FKBP1A in hepatocytes.
4.- The authors suggest that FKBP1A plays a carcinogenic role via increasing tumor immune cell infiltration and immune checkpoint genes expression (Discussion, page 472-475). To demonstrate this point, the authors should perform either FKBP1A “gain of function” or “loss of function” experiments.
Author Response
Comments and Suggestions for Authors
In this manuscript Li et al. have investigated the expression levels of FKBP1A and their diagnostic and prognostic value in liver hepatocellular carcinoma (LIHC). Based on the information provided by multiple on-line data bases, and some results from immunohistochemical analyses of commercially available human liver sample chips, the authors found that the expression of FKBP1A increased in the liver of LIHC patients, mainly localizing on the endoplasmic reticulum of hepatocytes. Moreover, they found that high expression of FKBP1A correlated with malignant tumor growth and poor survival of LIHC patients, being associated with infiltration of immune cells and expression of specific markers of M2 macrophages and immune checkpoint receptors, suggesting that FKBP1A could be a potential prognostic target involved in LIHC immune cell infiltration.
Author response to reviewer comments and suggestions: We thank the reviewer for this comment and suggestions on our study in our manuscript. They are all critical for us improve it.
Major comment: “The work by Li et al. represents an interesting approach to analyse the relationship between the immunophilin FKBP1A and liver hepatocellular carcinoma. Helped by detailed information from public databases, the authors have found a potential use of FKBP1A expression as prognosis factor to evaluate disease progression in LIHC patients. From this point of view, the work may be considered new and interesting. However, results are merely correlative and no insights about the possible role played by FKBP1A in LIHC are provided. In this regard, additional important questions remain. For example, whether FKBP1A is certainly involved in the mechanism of tumorigenesis. Additional experimental work should be thus required to demonstrate the authors’ suggestion that FKBP1A is involved in tumor immune cell infiltration in LIHC. Any evidence on this direction will considerably increase the manuscript’s interest and novelty.”
Author response to reviewer major comment: We are very grateful for his specific comments and valuable suggestions.
Indeed, we found a potential relevance of FKBP1A in LIHC based on the information provided by multiple on-line data bases, and some results from immunohistochemical analyses of commercially available human liver sample chips, particularly through the regulation of tumor cell immune infiltration. The involvement of FKBP1A in tumorigenesis has been reported, such as head and neck squamous cell carcinoma, lung adenocarcinoma, and breast cancers. In many more studies on in vitro cell culture system as well. mTOR signaling pathway could be the key link leading to tumorigenic pathology.
Therefore, we added the following in INTRODUCTION of our revision “Multiple studies reported that FKBP1A involved in tumorigenesis including head and neck squamous cell carcinoma (Ref. [19] in the revised manuscript), prostatic cancer cell line (Ref. [20] in the revised manuscript), lung adenocarcinoma (Ref. [21] in the revised manuscript) and breast cancer (Ref. [12] in the revised manuscript). In immune system, FKBP1A were directly led to mTOR activity via rapamycin recruited SIRT2 for its deacetylation (Ref. [22] in the revised manuscript). Recently, miR-195-5p/FKBP1A were identified as a key factor for paclitaxel-resistant prostate cancer cells to paclitaxel via LncRNA AFAP1-AS binding to the 3ʹUTR of FKBP1A (Ref. [20] in the revised manuscript).” (Page 2, Lines 40-45 in the revised manuscript). And the related references were cited in REFERENCES (Ref 12, 19-22).
According to the comments and suggestions of the reviewer, we carried out an additional important experiment on main calcium channel i.e. Ryanodine Receptor to which FKBP1A binds for regulation of ER membrane on calcium ion release from ER calcium pool. Localization of FKBP1A and RyR emphasize the critical role of FKBP1A protein. Additional texts were added in revised manuscript. Including RESULTS: “In addition, we also performed immunofluorescence staining for the endoplasmic reticulum (ER) marker ryanodine receptor (RyR) and co-localization analysis of FKBP1A with RyR (Figure 3A e and f in the revised manuscript).’’ (Page 5, Lines 21-23 in the revised manuscript), MATERIALS AND METHODS (Page 20, Lines 37-41 in the revised manuscript) and DISCUSSION (Page 15, Lines 50-52 in the revised manuscript). RyR IHCC image and its legend were added in Figure 3 with additional panel (Figure 3 A e and f in the revised manuscript) and its legend (Page 6, Lines 15-16 in the revised manuscript).
Some specific points
Reviewer comment 1.- Figure 3A-C: The number of microscopy fields that have been analysed and the number of controls and patients that have been studied should be indicated on the figure legend.
Author response to reviewer specific comment 1: We thank the reviewer for this specific comments. We have added the number of microscopy fields that have been analyzed and the number of controls and patients that had been studied on the figure legend (Page 6, Lines 17 in the revised manuscript).
Reviewer comment 2.- Figure 3B and 3C: Why is the background expression of FKBP1A (mean optical densities) so different between normal and LIHC immunohistochemistry? Specific values of FKBP1A expression in normal and LIHC samples should be normalized to their respective backgrounds.
Author response to reviewer specific comment 2: We thank the reviewer for this specific point very much. We showed a representative sample that in Figure 3A (panels a and b) that their backgrounds were slightly different as usual. Just like the reviewer point out, we DID normalized the values of normal and LIHC to their background respectively. In this revised manuscript, we re-calculated original data again and confirmed the result again: The expression level FKBP1A is higher in LIHC than that in normal livers. To terminate this possible misunderstanding, we combined panels B and C together as panel B (Page 6, Figure 3B in the revised manuscript).
Reviewer comment 3.- Figure 3D and E: The number of samples, donors and patients analysed in immunohistochemistry studies should be also indicated. On the other hand, immunofluorescence confocal microscopy analysis of co-localization between FKBP1A and an ER specific marker is required to characterize in depth the cellular localization of FKBP1A in hepatocytes.
Author response to reviewer specific comment 3: We thank the reviewer for the specific comments and suggestions. We have added the number of controls and patients that have been studied on the figure legend (Page 6, Lines 22 in the revised manuscript).
By following suggestion from the reviewer, we added an Endoplasmic Reticulum (ER) specific marker RyR to present ER in our revision (Page 6, Figure 3A e and f in the revised manuscript). ER distribution of RyR immunofluorescence is consistent with the ER distribution of FKBP1A. As this ER marker is the target protein of FKBP1A and RyR-FKBP1A complex is critical in cellular calcium signal, we did add relative information of RyR-FKBP1A in physiological normal and pathological tumor cells in INTRODUCTION: ‘‘As a member of immunophilin superfamily, FKBP1A can bind to immunosuppressive drugs including FK506, rapamycin and cyclosporin A (CsA) and interacts in mTOR pathway. FKBP1A was first identified in dendrite cells of peripheral blood cells and its related molecular mechanism had been under investigation for decades. One FKBP1A molecules bind to a RyR unite of RyR tetramer, and RyR–FKBP1A complex is in charge of excitation-contractile coupling in muscle, and excitation-conduction contractile coupling in neuronal cells via calcium-induced calcium release (CICR) signaling system (Ref. [14-16] in the revised manuscript). The FKBP1A-RyR complex medicated CICR plays important role in multiple cell biological process in physiological normal and pathological cells.” (Page 2, Lines 22-30 in the revised manuscript) and DISCUSSION: ‘‘Up-to-date, molecular mechanism of how FKBP1A functions in liver hepatocellular carcinoma remains unidentified and our data provide valuable clue that FKBP1A-RyR complex may be critically involved in pathological LIHC.’’ (Page 15, Lines 50-52 in the revised manuscript) of our revised manuscript.
Reviewer comment 4.- The authors suggest that FKBP1A plays a carcinogenic role via increasing tumor immune cell infiltration and immune checkpoint genes expression (Discussion, page 472-475). To demonstrate this point, the authors should perform either FKBP1A “gain of function” or “loss of function” experiments.
Author response to reviewer specific comment 4: We thank the reviewer for these precious comments and suggestions.
We understood that perform either FKBP1A “gain of function” or “loss of function” experiments would be useful to demonstrate our conclusions. Essentially, this also is one of our further plan that we are going to carry out within in vitro culture system to investigate deep molecular mechanism on RyR-FKBP1A and LIHC. Here in the present study, we mainly focused on the clinical relevance and significance of FKBP1A in predicting survival and progression by regulating immune cell infiltration. We will follow your suggestion and perform either FKBP1A “gain of function” and “loss of function” experiments in cell culture system.
As an importance for rational of this study, we added additional information of FKBP1A and RyR in the INTRODUCTION: ‘‘As a member of immunophilin superfamily, FKBP1A can bind to immunosuppressive drugs including FK506, rapamycin and cyclosporin A (CsA) and interacts in mTOR pathway. FKBP1A was first identified in dendrite cells of peripheral blood cells and its related molecular mechanism had been under investigation for decades. One FKBP1A molecules bind to a RyR unite of RyR tetramer, and RyR–FKBP1A complex is in charge of excitation-contractile coupling in muscle, and excitation-conduction contractile coupling in neuronal cells via calcium-induced calcium release (CICR) signaling system (Ref. [14-16] in the revised manuscript). The FKBP1A-RyR complex medicated CICR plays important role in multiple cell biological process in physiological normal and pathological cells.” (Page 2, Lines 22-30 in the revised manuscript)
We added additional information of FKBP1A and RyR in the INTRODUCTION: ‘‘Up-to-date, molecular mechanism of how FKBP1A functions in liver hepatocellular carcinoma remains unidentified and our data provide valuable clue that FKBP1A-RyR complex may be critically involved in pathological LIHC.’’ (Page 15, Lines 50-52 in the revised manuscript).
Actually, in published original articles of both animal models of FKBP1A “gain of function” or “loss of function”, the mice exhibited strong phenotypes on cardiac function (the senior author Dr. XU of this submission generated the cardiac specific animal models of MHC-FKBP1A “gain of function”). We summarized these information and added in DISCUSSION as “As FKBP1A were first identified in dendrite cells of peripheral blood, the convenient knockout mouse model of FKBP1A gene was generated with homologous recombination. The mice with FKBP1A deletion experienced severer cardiac defect of septal development without ventricular septum accomplishment and underwent early postnatal lethal as ventricular septal defect (VSD) symptoms in human. Because the septal tunnel unclosed with extra massive growth of ventricular tubercles in the mouse along with the syndrome and cranial neural tube closure defects, the animals cannot survive after birth (Ref. [26] in the revised manuscript). The cardiac specific overexpression of FKBP1A cDNA driven by MHC promoter was carried out in mouse as well. The cardiac phenotype caused by gain of FKBP1A gene experienced the critical sudden cardiac death with high-grade conduction system dysfunction leading to potassium channel disorder (Ref. [27] in the revised manuscript). Up-to-date, molecular mechanism of how FKBP1A functions in liver hepatocellular carcinoma remains unidentified and our data provide valuable clue that FKBP1A-RyR com-plex may be critically involved in pathological LIHC.” (Page 15, Lines 40-52 in the revised manuscript).
Reviewer 3 Report
The authors cover interesting topic regarding HCC.
Major:
1) The rationale of why the authors came up with this PROJECT.
2) What is the information that is not exactly available that motivated the authors to come up with this information. What are the current caveats and how do the authors highlight the current research in answering them? If not they need to address in future directions.
3) As is now well known, tumors grow and evolve through a constant crosstalk with the surrounding microenvironment, and emerging evidence indicates that angiogenesis and immunosuppression frequently occur simultaneously in response to this crosstalk.
4) Accordingly, strategies combining anti-angiogenic therapy and immunotherapy seem to have the potential to tip the balance of the tumor microenvironment and improve treatment response.
5) In the frame of points 3) and 4)thinking, HCC is one of most common cancers and the fourth leading cause of death worldwide. Commonly, HCC development occurs in a liver that is severely compromised by chronic injury or inflammation. Liver transplantation, hepatic resection, radiofrequency ablation (RFA), transcatheter arterial chemoembolization (TACE), and targeted therapies based on tyrosine protein kinase inhibitors are the most common treatments. The latter group have been used as the primary choice for a decade. However, tumor microenvironment in HCC is strongly immunosuppressive; thus, new treatment approaches for HCC remain necessary. The great expression of immune checkpoint molecules, such as programmed death-1 (PD-1), cytotoxic T-lymphocyte antigen 4 (CTLA-4), lymphocyte activating gene 3 protein (LAG-3), and mucin domain molecule 3 (TIM-3), on tumor and immune cells and the high levels of immunosuppressive cytokines induce T cell inhibition and represent one of the major mechanisms of HCC immune escape. Recently, immunotherapy based on the use of immune checkpoint inhibitors (ICIs), as single agents or in combination with kinase inhibitors, anti-angiogenic drugs, chemotherapeutic agents, and locoregional therapies, offers great promise in the treatment of HCC: please refer to PMID: 34065489 and expand
6) The authors need to highlight what new information the review is providing to enhance the research in progress.
7)The authors could provide a little more consideration of genomic directed stratifications in clinical trial design and enrollments.
8)The underlying message here is that more precision and individualized approaches need to be tested in well designed clinical trials – a challenge, but I would be interested in their perspective of how this might be done.
9) How did the authors quantify the IHC positivity?
10) Please add exact definition of X biomarker high/low (i.e. FKBP1A in “Statistical Analysis” section of “Methods”. Please include a figure visualizing absolute FKBP1A expression across the entire cohort
11) Did the authors check for hazard proportionality before proceeding with multivariable analysis?
Author Response
Major comments and Suggestions for Authors: The authors cover interesting topic regarding HCC.
Author response to reviewer comments and suggestions: We appreciate the insightful comments and suggestions from this reviewer. Via following your suggestions, we will greatly improve our manuscripts.
Major:
Reviewer major comment 1) The rationale of why the authors came up with this PROJECT.
Author response to reviewer major comment 1: We thank the reviewer for this comment and suggestion. We did add the rationale of this project in INTRIDUCTION as “As a member of immunophilin superfamily, FKBP1A can bind to immunosuppressive drugs including FK506, rapamycin and cyclosporin A (CsA) and interacts in mTOR pathway. FKBP1A was first identified in dendrite cells of peripheral blood cells and its related molecular mechanism had been under investigation for decades. One FKBP1A molecules bind to a RyR unite of RyR tetramer, and RyR–FKBP1A complex is in charge of excitation-contractile coupling in muscle, and excitation-conduction contractile coupling in neuronal cells via calcium-induced calcium release (CICR) signaling system (Ref. [14-16] in the revised manuscript). The FKBP1A-RyR complex medicated CICR plays important role in multiple cell biological process in physiological normal and pathological cells.” (Page 2, Lines 22-30 in the revised manuscript).
And more information for rationale added as “Multiple studies reported that FKBP1A involved in tumorigenesis including head and neck squamous cell carcinoma (Ref. [19] in the revised manuscript), prostatic cancer cell line (Ref. [20] in the revised manuscript), lung adenocarcinoma (Ref. [21] in the revised manuscript) and breast cancer (Ref. [12] in the revised manuscript). In immune system, FKBP1A were directly led to mTOR activity via rapamycin recruited SIRT2 for its deacetylation (Ref. [22] in the revised manuscript). Recently, miR-195-5p/FKBP1A were identified as a key factor for paclitaxel-resistant prostate cancer cells to paclitaxel via LncRNA AFAP1-AS binding to the 3ʹUTR of FKBP1A (Ref. [20] in the revised manuscript).” (Page 2, Lines 40-45 in the revised manuscript).
Reviewer major comment 2) What is the information that is not exactly available that motivated the authors to come up with this information. What are the current caveats and how do the authors highlight the current research in answering them? If not they need to address in future directions.
Author response to reviewer major comment 2: We thank the reviewer for this comment on our manuscript. We are so sorry to confuse the reviewer because we did not make it clear in our manuscript. In order to explain our current work more clearly, we have made corresponding changes in the revised manuscript and hope the results will be easier for the reader to understand (Page 2, Lines 31-39 in the revised manuscript): ‘’In recent years, emergence of immunotherapy has brought up strong hope into the treatment of liver cancer. Application of immune checkpoint inhibitors (ICIs), especially programmed cell death protein 1 (PD-1), programmed cell death ligand 1 (PD-L1), and cytotoxic T lymphocyte antigen 4 (CTLA-4) has shown great promise and critical progress for LIHC treatment (Ref. [17] in the revised manuscript). However, many cancer patients fail to respond to immune checkpoint blockade (Ref. [18] in the revised manuscript). Basic research and clinical trials exploring biomarkers of immunotherapy to predict efficacy are still limited, and it is not certain on which biomarkers can effectively examine the efficacy of immunotherapy (Ref. [18] in the revised manuscript). Our study on FKBP1A and LIHC may provide competent possible in pathological event.’’
To provide more comprehension and prospect for further study, we also added more information in DISCUSSION as “As FKBP1A were first identified in dendrite cells of peripheral blood, the convenient knockout mouse model of FKBP1A gene was generated with homologous recombination. The mice with FKBP1A deletion experienced severer cardiac defect of septal development without ventricular septum accomplishment and underwent early postnatal lethal as ventricular septal defect (VSD) symptoms in human. Because the septal tunnel unclosed with extra massive growth of ventricular tubercles in the mouse along with the syndrome and cranial neural tube closure defects, the animals cannot survive after birth (Ref. [26] in the revised manuscript). The cardiac specific overexpression of FKBP1A cDNA driven by MHC promoter was carried out in mouse as well. The cardiac phenotype caused by gain of FKBP1A gene experienced the critical sudden cardiac death with high-grade conduction system dysfunction leading to potassium channel disorder [(Ref. [27] in the revised manuscript). Up-to-date, molecular mechanism of how FKBP1A functions in liver hepatocellular carcinoma remains unidentified and our data provide valuable clue that FKBP1A-RyR com-plex may be critically involved in pathological LIHC.” (Page 15, Lines 40-52 in the revised manuscript)
Additionally, more references for our extra information added in revision were also cited in REFERENCES.
Reviewer major comment 3) As is now well known, tumors grow and evolve through a constant crosstalk with the surrounding microenvironment, and emerging evidence indicates that angiogenesis and immunosuppression frequently occur simultaneously in response to this crosstalk.
Author response to reviewer major comment 3: We thank this reviewer greatly for very professional comments. We strongly agree with the reviewer on these points. And also, we followed the Reviewer major comment 5) and added this incredible information in DISCUSSION. (Page 16, Lines 27-34; 38-39 in the revised manuscript).
Reviewer major comment 4) Accordingly, strategies combining anti-angiogenic therapy and immunotherapy seem to have the potential to tip the balance of the tumor microenvironment and improve treatment response.
Author response to reviewer major comment 4: We thank this reviewer greatly for very professional comments. We strongly agree with the reviewer on these points. And also, we followed the Reviewer major comment 5) and added this incredible information in DISCUSSION. (Page 16, Lines 27-34; 38-39 in the revised manuscript).
Reviewer major comment 5) In the frame of points 3) and 4) thinking, HCC is one of most common cancers and the fourth leading cause of death worldwide. Commonly, HCC development occurs in a liver that is severely compromised by chronic injury or inflammation. Liver transplantation, hepatic resection, radiofrequency ablation (RFA), transcatheter arterial chemoembolization (TACE), and targeted therapies based on tyrosine protein kinase inhibitors are the most common treatments. The latter group have been used as the primary choice for a decade. However, tumor microenvironment in HCC is strongly immunosuppressive; thus, new treatment approaches for HCC remain necessary. The great expression of immune checkpoint molecules, such as programmed death-1 (PD-1), cytotoxic T-lymphocyte antigen 4 (CTLA-4), lymphocyte activating gene 3 protein (LAG-3), and mucin domain molecule 3 (TIM-3), on tumor and immune cells and the high levels of immunosuppressive cytokines induce T cell inhibition and represent one of the major mechanisms of HCC immune escape. Recently, immunotherapy based on the use of immune checkpoint inhibitors (ICIs), as single agents or in combination with kinase inhibitors, anti-angiogenic drugs, chemotherapeutic agents, and locoregional therapies, offers great promise in the treatment of HCC: please refer to PMID: 34065489 and expand
Author response to reviewer major comment 5: We really appreciate the valuable comments and useful suggestions from the reviewer on our manuscript. We have brought this great point into the DISCUSSION, which can definitely increase the novelty of our manuscripts for readers as suggested by the reviewer. We also inserted the mentioned paper as a reference (Ref. [17]) and some expanding references in the revised manuscript, Ref. [34-37].
The extra DISCUSSION as added as “Regular treatments for LIHC include surgery, radiation therapy and targeted therapies are based on tyrosine protein kinase inhibitors (Ref. [17,34] in the revised manuscript). However, the prognosis of hepatocellular carcinoma is very poor due to drug resistance and frequent recurrence and metastasis. Recently, new therapeutic strategies such as immunosuppressive therapy for cancer depending on ICIs have shown very promising results (Ref. [17, 35-37] in the revised manuscript). The combination of conventional therapies and immunotherapy can achieve greater efficacy through further synergistic effects in LIHC (Ref. [17, 37] in the revised manuscript).” (Page 16, Lines 27-34 in the revised manuscript). And “Therefor, FKBP1A may serve as a potential target to increase the effectiveness of immunotherapy in LIHC.” (Page 16, Lines 38-39 in the revised manuscript).
Reviewer major comment 6) The authors need to highlight what new information the review is providing to enhance the research in progress.
Author response to reviewer major comment 6: We thank this reviewer greatly for kind comments. We appreciate this suggestion and we have modified it accordingly in the DISCUSSION section (Page 17, Lines 9-12 in the revised manuscript): “Our study provides recent evidence that FKBP1A is a potential prognostic factor in LIHC, since that FKBP1A may be involved in immune cell infiltration-related signaling path-ways to mediate LIHC development.”
Reviewer major comment 7) The authors could provide a little more consideration of genomic directed stratifications in clinical trial design and enrollments.
Author response to reviewer major comment 7: We thank this reviewer for this great comments. It is true that more genetic features may be associated with clinical outcomes for LIHC. We have added the following in the revised manuscript (Page17, Lines 13-25 in the revised manuscript): ‘‘The development and progression of hepatocellular carcinoma involves alterations in multiple signaling pathways. Each patient may respond differently to treatment modalities such as chemotherapy, radiation therapy, and immunotherapy. According to the cBioPortal databases, a few mutants occurred in exons encoding FKBP1A which all variations were not statistics related to LIHC. It seems the information embedded in the genomic introns should be investigated intensively for more comprehensive study. A current study linked LncRNA AFAP1-AS1 modulation and sensitivity to paclitaxel via miR-195-5p/FKBP1A axis (Ref. [20] in the revised manuscript) may give us clue to inspect mutation in genomic non-coding zone and its fundamental function in LIHC. Precision therapeutic approaches to cancer treatment, known as precision oncology, use the molecular characteristics of an individual patient's tumor to assess the likelihood of benefit or toxicity of a specific therapeutic intervention (Ref. [43] in the revised manuscript). It is imperative to search for specific tumor molecular markers, select sensitive populations, and fundamentally realize tailor-made individualized treatment.’’
Reviewer major comment 8) The underlying message here is that more precision and individualized approaches need to be tested in well-designed clinical trials – a challenge, but I would be interested in their perspective of how this might be done.
Author response to reviewer major comment 8: We thank this reviewer greatly for this valuable comment. We agree that more precision and individualized approaches need to be tested in well-designed clinical trials. It will be important to validate the clinical significance of FKBP1A and to investigate the potentiality of FKBP1A as a prognostic biomarker. We added extra information on molecular mechanism in FKBP1A-RyR complex in animal model of “gain of function” and “loss of function” in DISSCUSION as “As FKBP1A were first identified in dendrite cells of peripheral blood, the convenient knockout mouse model of FKBP1A gene was generated with homologous recombination. The mice with FKBP1A deletion experienced severer cardiac defect of septal development without ventricular septum accomplishment and underwent early postnatal lethal as ventricular septal defect (VSD) symptoms in human. Because the septal tunnel unclosed with extra massive growth of ventricular tubercles in the mouse along with the syndrome and cranial neural tube closure defects, the animals cannot survive after birth (Ref. [26] in the revised manuscript). The cardiac specific overexpression of FKBP1A cDNA driven by MHC promoter was carried out in mouse as well. The cardiac phenotype caused by gain of FKBP1A gene experienced the critical sudden cardiac death with high-grade conduction system dysfunction leading to potassium channel disorder (Ref. [27] in the revised manuscript). Up-to-date, molecular mechanism of how FKBP1A functions in liver hepatocellular carcinoma remains unidentified and our data provide valuable clue that FKBP1A-RyR com-plex may be critically involved in pathological LIHC”. (Page 15, Lines 40-52 in the revised manuscript). This additional discussion may provide more comprehension and prospect for further study of our readers and ourselves.
We also input suggested approach for further well-designed clinical trials in DISCUSSION as the following “According to our data, the increase expression of FKBP1A in dendrite cells and M2 mac-rophages of peripheral blood in LIHC. It could be possible that collecting of and evaluating the FKBP1A expression in patients’ peripheral blood may be practical approach for clinical therapeutic diagnostics.” (Page 16, Lines 19-22 in the revised manuscript).
We have modified it accordingly in the results section (Page 17, Lines 27-30 in the revised manuscript): “More precision and individualized approaches need to be tested in well-designed clinical trials. It will be important to validate the clinical significance of FKBP1A and to investigate the potentiality of FKBP1A as a prognostic biomarker in the future.’’
Reviewer major comment 9) How did the authors quantify the IHC positivity?
Author response to reviewer major comment 9: We thank this reviewer for this specific comments. We quantified the immunohistochemical positivity of FKBP1A by referring to this literature (Ref. [57] in the revised manuscript). We have added the following in “Statistical Analysis of IHC” section of “Materials and Methods” (Page 19, Lines 44-45 in the revised manuscript): “Quantitative assessment of IHC images of human tissue samples for FKBP1A by IHC Profiler plugin of image J.’’
Reviewer major comment 10) Please add exact definition of X biomarker high/low (i.e., FKBP1A in “Statistical Analysis” section of “Methods”. Please include a figure visualizing absolute FKBP1A expression across the entire cohort.
Author response to reviewer major comment 10: We thank this reviewer greatly for very professional suggestion. In the IHC staining of entire cohort samples, we used the mean optical density (mean IOD) of FKBP1A positive staining as statistical analysis. We compared the expression levels between LIHC samples, normal liver tissues, and adjacent to LIHC tissues without defining high/low FKBP1A expression in the tissue samples. Because the Bioaitech Company who provided the tissue chip for our TMA analysis has no relevant clinical survival information to provide to us, we couldn’t carry out Statistical Analysis on high/low FKBP1A expression for survival analysis for the entire cohort.
Reviewer major comment 11) Did the authors check for hazard proportionality before proceeding with multivariable analysis?
Author response to reviewer major comment 11: We thank this reviewer greatly for very professional and careful reviewing our manuscript. We appreciate this suggestion and add the following statement (Page 8, Lines 48-54 in the revised manuscript) and a table (Table 2; Page 10, Lines 2-4 in the revised manuscript) in the RESULT: ‘‘To determine the risk factors related with LIHC overall survival, we used both univariate and multivariate Cox regression to carry out the analysis. Univariate Cox analysis identified the potential OS-related variables on FKBP1A and others including the age, gen-der, pT−stage, pTNM−stage and grade. The univariate (hazard ratio, 1.766; 95% CI, 1.375-2.268; p < 0.001; Table 2) and multi-variate (hazard ratio, 1.546; 95% CI, 1.185-2.017; p = 0.001; Table 2) Cox analyses showed that FKBP1A was an independent prognostic risk factor for LIHC overall survival (Table 2).’’
Also, we have added the relevant analysis methods in “Kaplan-Meier Plotter Analysis” section of “Materials and Methods” as the following “The relative prognostic value of the FKBP1A with that of routine clinicopathological features in the TCGA was evaluated with univariate/multivariate Cox proportional hazards analysis.” (Page 18, Lines 35-37 in the revised manuscript)
Table 2. Prognostic factors for overall survival in univariable and multivariable Cox proportional hazards analyses (Page 10, Lines 2-4 in the revised manuscript).
Factor |
Univariable Analysis |
Multivariable Analysis |
||
Hazard Ratio (95% CI) |
p-Value |
Hazard Ratio (95% CI) |
p-Value |
|
FKBP1A |
1.766 (1.375-2.268) |
< 0.001 |
1.546 (1.185-2.017) |
0.001 |
Age |
1.012 (0.999-1.026) |
0.078 |
1.008 (0.994-1.022) |
0.261 |
Gender |
0.816 (0.573-1.163) |
0.260 |
0.907 (0.623-1.320) |
0.611 |
pT−stage |
1.675 (1.397-2.007) |
< 0.001 |
1.804 (1.411-2.306) |
< 0.001 |
pTNM−stage |
1.376 (1.145-1.654) |
< 0.001 |
0.870 (0.687-1.101) |
0.246 |
Grade |
1.121 (0.887-1.417) |
0.339 |
1.079 (0.836-1.392) |
0.559 |
Round 2
Reviewer 2 Report
The manuscript by Li et al. has been considerably improved. However an important issue remains. Conversely to the authors´ statement: "we also performed ... co-localization analysis of FKBP1A 22 with RyR (Figure 3A e and f)", no co-localization analysis between these proteins is shown on figures 3Ae and 3Af, but rather a single staining of RyR (green). To actually perform their co-localization analysis, simultaneous two colour immunofluorescence staining of FKBP1A22 and RyR should be performed in control and LIHC samples.
Author Response
Comments and Suggestions for Authors
The manuscript by Li et al. has been considerably improved. However, an important issue remains. Conversely to the authors´ statement: "we also performed ... co-localization analysis of FKBP1A 22 with RyR (Figure 3A e and f)", no co-localization analysis between these proteins is shown on figures 3Ae and 3Af, but rather a single staining of RyR (green). To actually perform their co-localization analysis, simultaneous two colour immunofluorescence staining of FKBP1A22 and RyR should be performed in control and LIHC samples.
Author response to reviewer major comment: We are very grateful for his specific comments and valuable suggestions.
According to the comments and suggestions of the reviewer, we performed FKBP1A and Endoplasmic Reticulum (ER) specific marker RyR co-localization analysis. We used two colour immunofluorescence to check FKBP1A (red) and RyR (green) at the same time in control and LIHC samples. We made some changes on FIGURE 3 (Page 6, Figure 3 in the revised manuscript) by putting the alalysis of the FKBP1A and RyR co-localization on a new figure and added a new figure to show the co-localization of FKBP1A with the ER marker RyR in the RESULT (Page 7, Figure 4 in the revised manuscript). Its legend was added as well in the text of this revision (Page 7, Lines 5-15 in the revised manuscript). The step of co- immunofluorescence staining was added to the MATERIALS AND METHODS (Page 20, Lines 37-43 in the revised manuscript).
Reviewer 3 Report
I am satisfied with the answer provided.
Author Response
Comments and Suggestions for Authors
I am satisfied with the answer provided.
Author response to reviewer comments and suggestions: We thank the reviewer for taking efforts and spending time to review our manuscript. Our manuscripts have been greatly improved.